# THINK BEFORE YOU SPEAK:
# TRAINING LANGUAGE MODELS WITH PAUSE TOKENS

**Sachin Goyal**[*][†]
Machine Learning Department
Carnegie Mellon University
sachingo@andrew.cmu.edu

**Ziwei Ji**
Google Research, NY
ziweiji@google.com

**Ankit Singh Rawat**
Google Research, NY
ankitsrawat@google.com

**Aditya Krishna Menon**
Google Research, NY
adityakmenon@google.com

**Sanjiv Kumar**
Google Research, NY
sanjivk@google.com

**Vaishnavh Nagarajan**[†]
Google Research, NY
vaishnavh@google.com

## ABSTRACT

Transformer-based language models generate responses by producing a series of tokens in immediate succession: the $(K+1)^{\text{th}}$ token is an outcome of manipulating $K$ hidden vectors per layer, one vector per preceding token. What if instead we were to let the model manipulate say, $K + 10$ hidden vectors, before it outputs the $(K+1)^{\text{th}}$ token? We operationalize this idea by performing training and inference on language models with a (learnable) *pause* token, a sequence of which is appended to the input prefix. We then delay extracting the model's outputs until the last pause token is seen, thereby allowing the model to process extra computation before committing to an answer. We empirically evaluate *pause-training* on decoder-only models of 1B and 130M parameters with causal pretraining on C4, and on downstream tasks covering reasoning, question-answering, general understanding and fact recall. Our main finding is that inference-time delays show gains on our tasks when the model is both pretrained and finetuned with delays. For the 1B model, we witness gains on eight tasks, most prominently, a gain of $18\%$ EM score on the QA task of SQuAD, $8\%$ on CommonSenseQA and $1\%$ accuracy on the reasoning task of GSM8k. Our work raises a range of conceptual and practical future research questions on making delayed next-token prediction a widely applicable new paradigm.

## 1 INTRODUCTION

Transformer-based causal language models generate tokens one after the other in immediate succession. To generate the $(K+1)^{\text{th}}$ token, the model consumes the $K$ previous tokens, and proceeds layer by layer, computing $K$ intermediate vectors in each hidden layer. Each vector in itself is the output of a module (consisting of self-attention and multi-layer-perceptrons) operating on the previous layer's output vectors. However sophisticated this end-to-end process may be, it abides by a peculiar constraint: the number of operations determining the next token is limited by the number of tokens seen so far. Arguably, this was the most natural design choice when the Transformer was first conceived by Vaswani et al. (2017). But in hindsight, one may wonder whether for some inputs, the $(K+1)^{\text{th}}$ token demands $K + M$ Transformer operations in each layer (for $M > 0$), which cannot be met by the arbitrarily constrained $K$ operations per layer. This paper explores one way to free the Transformer of this arbitrary per-layer computational constraint.

The approach we study is to append dummy tokens into a decoder-only model's input, thereby *delaying* the model's output. Specifically, we select a (learnable) pause token (denoted <pause>) and append one or more copies of <pause> as a sequence to the input. We simply ignore the model's corresponding outputs until the last <pause> token is seen, after which we begin extracting its response.

---

[*]Work done in part as a Student Researcher at Google.
[†]Corresponding Authors

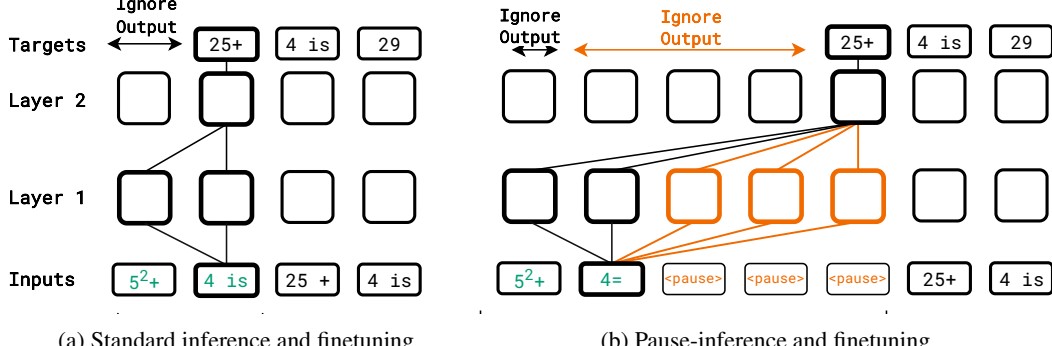

(a) Standard inference and finetuning        (b) Pause-inference and finetuning

Figure 1: **Standard vs. pause-inference (and finetuning)**. We consider a downstream task where, given a prefix, the decoder-only model (bidirectionally) attends to all of the prefix to generate its target answer. The rounded squares denote one Transformer operation (a self-attention and MLP) in a 2-layer Transformer. Any **Ignore Output** denotes that during inference, the corresponding output token is not extracted and thus, not fed back autoregressively; during finetuning, this output is not backpropagated through. The connecting lines denote some (not all) of the "computational pathways" within the model. Specifically, we visualize only those pathways that begin at a specific token in the prefix (here arbitrarily chosen to be "4 is") and end at an output token (here arbitrarily chosen to be "25+"). All differences between the two settings are highlighted in color. (a) In standard inference (finetuning), the model's output is extracted immediately upon seeing the last prefix token. (b) In pause-inference (and pause-finetuning), this is initiated only after appending a manually specified number of <pause> tokens. This introduces new computational pathways (the colored lines) between the prefix token and the output token of interest.

Crucially, we consider injecting such delays not just at inference, but also during downstream finetuning (see Fig 1) and pretraining (see Fig 2, which provides additional technical details).

*A-priori*, it is unclear what this simple change would bring about in practice. Optimistically, the Transformer may take advantage of a "wider" computational pathway induced by the delay. A more mundane outcome though would be that the model simply skips any delays introduced by the <pause> tokens. After all, neither do the <pause> tokens provide any additional information during inference, *nor are there sufficiently many new parameters* (barring the few embedding parameters of the single <pause> token) that can encode any additional information from training data. Worse still, these uninformative tokens may drown out informative signals, and hurt the model.

Partial answers to this question can be found in the literature, motivated somewhat differently. To understand where the benefits of chain-of-thought (Wei et al., 2022) come from, Lanham et al. (2023) append dummy thoughts in the form of periods ('...'), but only during inference. This, they report, does not help. Presumably, an off-the-shelf model may not have *learned* to utilize the new computational pathways offered by the inference-time delay. Burtsev et al. (2020) learn with *prepended* dummy tokens, with the orthogonal motivation of adding memory (rather than extending computation). They train with these tokens only on the target task, and observe minimal performance gains.

What then can we hope for when injecting (appended) delays on all stages of training and inference? Our work empirically evaluates this, and other key questions that come up when training the Transformer with delays. For this, we study pause-training on a 1B and 130M parameter decoder-only model, trained on C4 (Raffel et al., 2020) and finetuned on nine downstream tasks spanning extractive question answering, reasoning, general understanding and fact recall. In summary, we make the following contributions:

(1) We pose the question of *what happens if we delay a model's answer generation, and how can we execute these delays?* We design one way: training with dummy <pause> tokens. Accordingly, we design a pause-injected pretraining, downstream finetuning, and inference procedure.

(2) We find that on a variety of downstream tasks, training models with <pause> tokens during both pretraining and downstream finetuning, exhibits clear gains compared to standard end-to-end training and inference. Most notably, for the 1B model, in the SQuAD extractive question-

answering task, this approach improves the exact match score by $18\%$. Similarly we observe $8\%$ gains on the general understanding task of CommonSense QA and $1\%$ accuracy gain on the reasoning task of GSM8k over the standard model's accuracy of $7.5\%$.

(3) On the flip side, when we introduce `<pause>` tokens only in the downstream finetuning stage (on the standard pretrained model), we find that the gains show up in far fewer instances, and are relatively mild. In some instances, we even find a clear drop in performance.

(4) We also conduct a series of key ablations: (a) We find that *appending* `<pause>` tokens is largely better than *prepending* them, (b) perhaps unsurprisingly, for any downstream task, there is a corresponding optimal number of `<pause>` tokens, and (c) when decreasing the number of inference-time `<pause>` tokens, we find a graceful degradation of performance even though pause-training does not explicitly train for such robustness.

Overall, our work explores the new paradigm of delayed next-token generation in Transformer models, and finds that there *are* benefits to this simple change, provided the change is implemented both during pretraining and finetuning. Our preliminary step here inspires a variety of conceptual and practical future research questions, ranging from understanding how Transformer delays work mechanistically, to making pause-training more generally applicable for practice.

## 2   PRELIMINARIES

We briefly outline the next-token prediction process in a standard causal decoder-only language model (details in §A). Consider a vocabulary $\mathcal{V}$ and an input $\mathbf{p}_{1:K} \in \mathcal{V}^K$ of $K$ tokens. Let $f$ denote a Transformer-based language model, from which we sample the next token as $p_{K+1} \sim f(\mathbf{p}_{1:K})$. To achieve this, internally, each layer $l \in [1, L]$ of the Transformer produces an intermediate vector $\mathbf{v}_k^{(l)}$ corresponding to each input token. The next token, i.e., $p_{K+1}$ is then sampled from a distribution inferred from the last vector in the last layer, $\mathbf{v}_K^{(L)}$.

On a high level, each layer in the above process can be represented as a function $T : \mathbb{R}^{D \times K} \to \mathbb{R}^{D \times K}$. Its input is a matrix of $K$ vectors, $V_{1:K} = [\mathbf{v}_1, \ldots, \mathbf{v}_K]$, and likewise, the output, $V'_{1:K}$. This mapping itself involves two key (parameterized) modules. The first is the attention module $\Phi_{\text{Attn}}$ which takes as inputs two matrices $V_{\text{key}}, V_{\text{value}} \in \mathbb{R}^{D \times N}$ (for any $N$) and a "query" vector $\mathbf{v} \in \mathbb{R}^D$ to produce an output vector in $\mathbb{R}^D$. This is followed by a feedforward module $f_{\text{FF}} : \mathbb{R}^D \to \mathbb{R}^D$. Then, given the inputs $\mathbf{v}_k$, and given the layer-norm module $\Phi_{\text{LN}} : \mathbb{R}^D \to \mathbb{R}^D$, the outputs $\mathbf{v}'_k$ for $k \leq K$ can be expressed as:

$$\mathbf{a}_k = \Phi_{\text{LN}}\left(\Phi_{\text{Attn}}(V_{1:k}, V_{1:k}, \mathbf{v}_k) + \mathbf{v}_k\right) \tag{1}$$

$$\mathbf{v}'_k = \Phi_{\text{LN}}\left(\Phi_{\text{FF}}(\mathbf{a}_k) + \mathbf{a}_k\right). \tag{2}$$

Observe here that the $k^{\text{th}}$ output $\mathbf{v}'_k$ is obtained by manipulating exactly the $k$ previous hidden embeddings in the same layer, $V_{1:k}$.

## 3   PAUSE-TRAINING

In the current paradigm of language models, we compute exactly $K$ embeddings $\mathbf{v}_1, \ldots \mathbf{v}_K$ in each layer, before generating the $(K+1)^{\text{th}}$ token, $p_{K+1}$. Our premise is that this limit of $K$ operations is an arbitrary one. Instead, we wish to expend more than $K$ operations towards producing the next token, $p_{K+1}$. While something to this effect could be achieved by increasing the number of attention heads in each layer, we are interested in an orthogonal approach that introduces hardly any parameters into the network. The idea is to synthetically increase the input sequence length by appending $M$ dummy tokens to the input, thus delaying the model's next response by $M$ tokens of input. In effect, this $M$-token-delay lets the model manipulate an additional set of $M$ intermediate vectors $\mathbf{v}_{K+1}, \ldots, \mathbf{v}_{K+M}$ before committing to its next (output) token, $p_{K+1}$. Intuitively, these vectors could provide a richer representation of the input (e.g., by attending differently), thus resulting in a better next token from the model. We visualize this in Figure 1.

### 3.1   LEARNING AND INFERENCE WITH THE `<pause>` TOKEN

A simple choice for the dummy tokens are special characters such as '.' or '#', as Lanham et al. (2023) chose for inference. But to prevent the model from confounding the role of delays with the

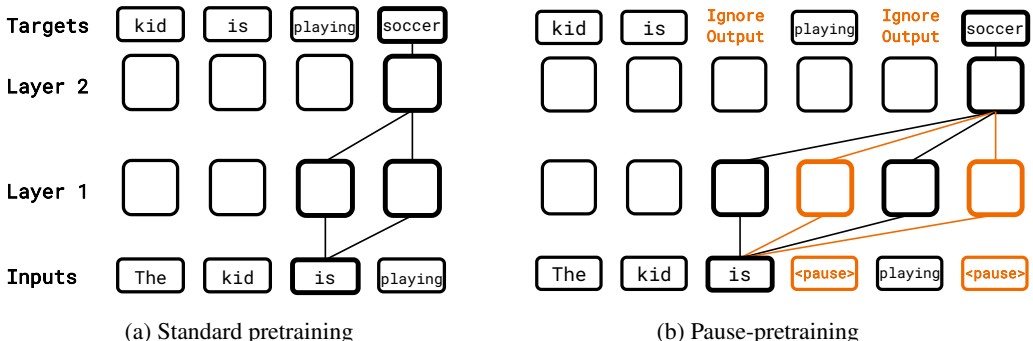

(a) Standard pretraining  (b) Pause-pretraining

Figure 2: **Standard vs. pause-pretraining**. We consider pretraining based on causal language modeling, where each token is predicted given all preceding tokens in the sequence, using unidirectional self-attention. Here, we visualize the computational pathways beginning from the token "is" on the input side of the decoder-only model, to a subsequent token "soccer" on the output side. Please see Figure 1 for a guide on how to follow this visualization. (a) In standard pretraining, we compute the model's loss at each output token, and backpropagate through it. (b) In pause-pretraining, we insert multiple copies of <pause> tokens at uniformly random locations in the input. However, we do not apply a loss on the model to predict these tokens, as indicated by each corresponding **Ignore Output** flags. This introduces new computational pathways connecting the input token and the output token of interest.

role the above characters play in natural language, we choose a *single* <pause> token residing outside of the standard vocabulary. To impose multi-token delays, we simply repeat this token. Building on this core idea, below we discuss our specific techniques for pause-pretraining and pause-finetuning.

**Pretraining with the <pause> token**   The sequences in our pretraining data do not come with an annotation of which suffix constitutes the answer, since every input token also functions as a target output. Thus, it is impossible to execute the simple delaying strategy of appending dummy tokens before extracting the answer. Therefore, for a given pretraining sequence $\mathbf{p}_{1:N}$, we insert multiple <pause> tokens (say $M_{\mathrm{pt}}$ many) at uniformly *random* locations to obtain a pause-injected sequence, $\tilde{\mathbf{p}}_{1:N+M_{\mathrm{pt}}}$. We visualize this in Figure 2b. We then train the model with the standard next-token prediction loss on this pause-injected sequence, while ignoring any loss term that corresponds to predicting the pause tokens themselves. Formally, let $S_{\mathrm{ignore}} = \{k\colon \tilde{p}_{k+1} = \text{<pause>}\}$ denote the positions where the *next* token is a <pause> token. Then, for the decoder-only language model $f$, the pause-training loss is given by:

$$\mathcal{L}_{\mathrm{PausePT}}(f, \tilde{\mathbf{p}}_{1:N+M_{\mathrm{pt}}}) = \sum_{\substack{k=1 \\ k \notin S_{\mathrm{ignore}}}}^{N+M_{\mathrm{pt}}-1} \mathcal{L}_{\mathrm{CE}}(\tilde{p}_{k+1}, f(\tilde{\mathbf{p}}_{1:k})), \tag{3}$$

where $\mathcal{L}_{\mathrm{CE}}$ denotes the cross-entropy loss. Observe that the loss is skipped over indices in $S_{\mathrm{ignore}}$. The rationale is that, we only want to use the <pause> tokens as a way of enforcing a delay in the model's computation; demanding that the model itself produce these tokens, would only be a pointless distraction. Finally, as is standard, we update the parameters of both the model and of all the tokens, including those of the <pause> token. We term this *pause-pretraining* (Algorithm 1).

**Finetuning with the <pause> token**   In downstream finetuning, we are given a prefix $\mathbf{p}_{1:N}$ annotated with a target $\mathbf{t}_{1:T}$. Here, we append $M_{\mathrm{ft}}$ copies of the <pause> token to $\mathbf{p}_{1:N}$, to create our new prefix, $\tilde{\mathbf{p}}_{1:N+M_{\mathrm{ft}}}$. We visualize how this introduces new computational pathways in Figure 1. As before, we ignore the model's outputs until the last <pause> token is seen. We apply the standard next-token prediction loss on the target with the new prefix, thus minimizing $\sum_{k=0}^{T-1} \mathcal{L}_{\mathrm{CE}}(t_{k+1}, f([\mathbf{p}_{1:N+M_{\mathrm{ft}}}, \mathbf{t}_{1:k}]))$, where $[\cdot]$ denotes the concatenation operation. Note that for any given downstream task, we fix $M_{\mathrm{ft}}$ to be the same across all inputs for that task. We again

update both the parameters of the model, and that of the whole vocabulary, including the `<pause>` token, as is standard. We term this *pause-finetuning* (Algorithm 2).

**Pausing during inference** During inference on the downstream task, we append $M_{\text{inf}}$ `<pause>` tokens to the prefix and as always, we ignore the output of the model until the last `<pause>` token (Figure 1). We term this *pause-inference* (Algorithm 3).

## 3.2 VARIANTS OF PAUSE-TRAINING

While pause tokens can be incorporated in either pretraining or finetuning, in our study, we will consider all combinations of this. Our hope is to identify if there are any differences in how each stage of pause-training affects inference-time performance. In total, we study four techniques:

1. Standard Pretraining and Standard Finetuning (**StdPT_StdFT**).
2. Standard Pretraining and Pause-Finetuning (**StdPT_PauseFT**): We train with `<pause>` tokens only during downstream finetuning. *If* this technique helps, it would promise a practically viable approach for pause-training off-the-shelf models.
3. Pause-Pretraining and Standard Finetuning (**PausePT_StdFT**): Here we introduce `<pause>` tokens during pretraining, but abandon it downstream. This is purely for analytical purposes (See §4.3).
4. Pause-Pretraining and Pause-Finetuning (**PausePT_PauseFT**): We inject delays into both stages.

Unless stated otherwise, we use the same number of pauses at inference as finetuning ($M_{\text{inf}} = M_{\text{ft}}$).

## 4 EXPERIMENTS

Our main experiments broadly aim to address two questions:

(1) Does delaying the model computation via pausing help (hopefully, due to the wider computational flow), have no effect (since the tokens provide no new information, and substantially no new parameters are added) or hurt (perhaps, by distracting the model with stray tokens)?

(2) If at all these delays have any effect, is there a difference in performance when we inject it into the pretraining stage versus finetuning stage versus both?

## 4.1 EXPERIMENT SETUP

We consider decoder-only models of size 1B and 130M for our main experiments. For our ablations, we stick to the 1B model. Both the standard and pause models are pretrained on the C4 English mixture (Raffel et al., 2020), using the causal next token prediction objective for a total of 200B tokens (slightly more than 1 epoch on C4). For pause-pretraining, we insert the `<pause>` token randomly at 10% of the sequence length (2048) positions, and trim the now-longer sequence to its original sequence length. We then conduct pause-pretraining and standard-pretraining for the same number of total tokens (200B). We use a *single* `<pause>` token embedding, effectively increasing the parameter count by 1024 (the token embedding size), a quantity that is dwarfed by the 1 billion total parameter count (the token constitutes a $10^{-6}$ fraction of the model size).

Since we expect different downstream tasks to benefit from a different number of finetuning `<pause>` tokens $M_{\text{ft}}$, we run finetuning with $M_{\text{ft}}$ (and likewise $M_{\text{inf}}$) set to 10 and 50 and report the best of these two for our consolidated results. However, we provide the values for both $M_{\text{ft}} \in \{10, 50\}$ in Appendix E, in addition to a more finegrained ablation of this hyperparameter in Section 5. For all the downstream finetuning experiments, we report mean and standard deviation over 5 runs (with the randomness purely from the finetuning stage). We tune the learning rate and batch size for standard end-to-end training, and use the best hyperparameter for all other training variants as well. We share all the hyperparameters in Appendix H.

## 4.2 DOWNSTREAM DATASETS

We consider nine varied downstream tasks: (a) reasoning (GSM8k (Cobbe et al., 2021)), (b) extractive question answering (SQuAD (Rajpurkar et al., 2016), CoQA (Reddy et al., 2019)), (c) general understanding (CommonSenseQA (Talmor et al., 2019), PhysicalIQA (Bisk et al., 2020)), (d)

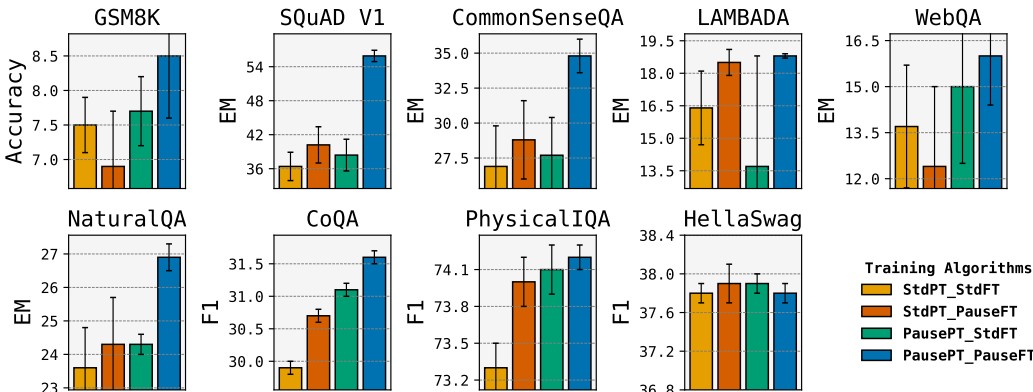

Figure 3: **Downstream performance for a 1B model.** Injecting delays in both stages of training (`PausePT_PauseFT`) outperforms the standard end-end training `StdPT_StdFT` on our wide variety of tasks (except HellaSwag). In contrast, introducing delays only in the finetuning stage provides only lukewarm gains, and even hurts in GSM8k.

long term context recall (LAMBADA (Paperno et al., 2016)), (e) natural language inference (HellaSwag (Zellers et al., 2019)), and (f) fact recall (WebQuestions (Berant et al., 2013), Natural Questions (Kwiatkowski et al., 2019)). HellaSwag and PhysicalIQA are scoring tasks. We note that our implementation of CommonSenseQA is as a decoding task, and hence we report Exact Match (EM) scores. Detailed dataset description is in Appendix G.

### 4.3 RESULTS: EFFECT OF PAUSE-TRAINING

We report the performance of the four approaches in §3.2 on all our downstream tasks for our 1B model in Figure 3, and for our 130M model in Appendix B. We discuss zero-shot results in §D.

**The benefit of pause-pretraining followed by pause-finetuning (`PausePT_PauseFT`).** Our first core finding is that there are clear gains when `<pause>` tokens are introduced during both pretraining and finetuning (`PausePT_PauseFT`), across a majority of the tasks we consider. Overall, this outperforms the standard baseline (`StdPT_StdFT`) on eight tasks for the 1B model, and on six tasks for the 130M model (Appendix Fig 5) albeit to varying extents. Most prominently, for the 1B model on the SQuAD question-answering task, `PausePT_PauseFT` improves over `StdPT_StdFT` by an $18\%$ EM score. Similarly, we observe upto $8\%$ gains on the general understanding task of CommonSenseQA. On the reasoning task of GSM8k, `PausePT_PauseFT` gives an accuracy of $8.5\%$ compared to $7.5\%$ of the standard baseline. Similar gains are observed in other tasks like long-term context understanding (LAMBADA) and also on fact recall tasks like WebQA and NaturalQuestion.

**The lukewarm effect of pause-finetuning a standard-pretrained model (`StdPT_PauseFT`).** In contrast to the above observations, introducing delays only during downstream finetuning (`StdPT_PauseFT`) gives mixed results. While there are gains on about 5 benchmarks, they are comparitively less. On the remaining, the performance mirrors (or is worse than) standard training.

**Isolating the benefits of pause-pretraining independent of downstream delay (`PausePT_StdFT`).** The gains in `PausePT_PauseFT` may come not only from inference-time delays, but also from better representations learned by pause-pretraining. To isolate the latter effect, we consider `PausePT_StdFT`, where we do not inject delays in the downstream task. Here the gains are clear only in two tasks (CoQA and PhysicalIQA). Thus, we conclude that pause-pretraining improves the representation for a few downstream tasks; conversely, in most tasks, the gains of `PausePT_PauseFT` must come from well-learned delayed computations executed at inference-time.

**Filler characters as `<pause>`:** For completeness, we also report results for inference on `StdPT_StdFT` models, delayed with $10$ or $50$ periods ('.'). Corroborating the observations of Lanham et al. (2023), we find no gains in doing this (Figure 4a).

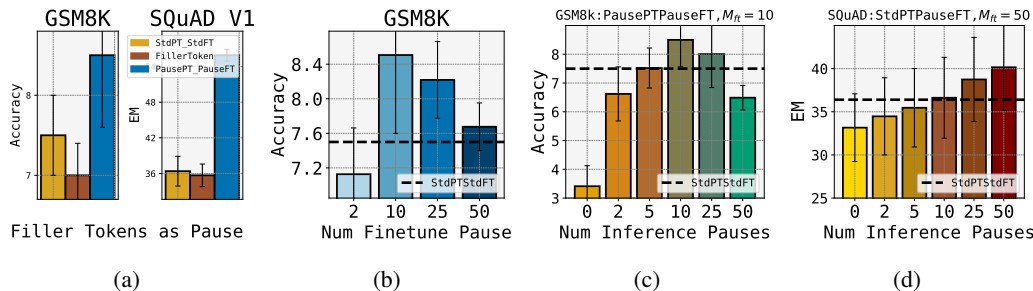

Figure 4: **Key Ablations (§5):** (a) We compare a pause-*trained* model vs. a standard model delayed using a filler periods ('...'). As in Lanham et al. (2023), the filler periods do not give any gains. (b) There exists an optimal number of finetuning <pause> tokens ($M_{\text{ft}}$) for a given downstream dataset beyond which gains diminish. (c) and (d) We test the robustness of pause-trained models to varying number of inference time <pause> tokens (setting $M_{\text{inf}}$ not equal to $M_{\text{ft}}$), which exposes the model to a serious test-time distribution shift. Pause-training degrades gracefully to shifts as wide as $M_{\text{inf}} \in [5, 25]$ for $M_{\text{ft}} = 10$ both for (c) PausePT_PauseFT and (d) StdPT_PauseFT.

Thus, to the core question of our exploration — whether delays help, hurt or do nothing at all — we find that the answer depends on when these delays are introduced. Specifically, pause-*pre*training appears crucial for delays to help in downstream inference-time. We conjecture that a standard-pretrained model has strong biases that prevent it from fully realizing the benefits of inference-time delays e.g., standard pretraining biases the model to be "quick" in its computations.

**Remark:** As a concluding note, we remind the reader that the PausePT_PauseFT model has a (deliberately injected) computational advantage compared to StdPT_StdFT, during finetuning and inference. However, there is no computational advantage during pause-pretraining since we equalize the number of tokens seen. In fact, this only results in a slight statistical disadvantage: the pause-pretrained model sees only 90% of the (meaningful) pretraining tokens that the standard model sees, as the remaining 10% are dummy <pause> tokens.

## 5 ABLATIONS: WHERE AND HOW MANY <pause> TOKENS TO USE

NUMBER OF <pause> TOKENS DURING FINETUNING Recall that we append $M_{\text{ft}}$ copies of (the same) <pause> token to the prefix during finetuning. We find that for each downstream dataset, there is a corresponding optimal value of $M_{\text{ft}}$. For example, on GSM8k, 10 <pause> tokens are optimal with accuracy reducing to that of baseline as <pause> tokens are increased to 50 (See Figure 4b), while for SQuAD , 10 is sub-optimal (see Appendix E). Possibly, for each dataset, there exists a certain threshold of <pause> tokens beyond which the self-attention mechanism becomes overwhelmed.

ROBUSTNESS TO A VARYING NUMBER OF INFERENCE-TIME PAUSES So far, we have set the inference-time delay to be the same as what was seen during finetuning ($M_{\text{inf}} = M_{\text{ft}}$). Next, we examine what happens if we vary $M_{\text{inf}}$ during inference. Note that this presents a severe test-time distribution shift as we provide no supervision for the model until the last <pause> token (the $M_{\text{ft}}^{\text{th}}$ one) is seen. Thus the model may very well output garbage if we begin eliciting a response that is either premature ($M_{\text{inf}} < M_{\text{ft}}$) or belated ($M_{\text{inf}} > M_{\text{ft}}$). Yet, in practice, we find a (somewhat) graceful behavior.

First, PausePT_PauseFT model is robust to a wide test-time shift in the number of <pause> tokens (see Figure 4c and Appendix F): the performance remains above the baseline even if the inference-time pause tokens are half that of training-time. This is desirable in case of real-time fluctuations in computational constraints. Relatively, the StdPT_PauseFT model (wherein we inject delays only during finetuning) is even more more robust (see Fig 4d and also Appendix F).

Now, an ideal robustness criterion would be that, in the absence of *any* <pause> tokens, the pause-finetuned model performs just as well as a standard-finetuned model. Unfortunately, this isn't the case for any of our models. In fact, for PausePT_PauseFT, providing zero delay during inference

breaks its performance spectacularly (Figure 4c and also Appendix F), even if the model behaves reasonably with as few as 2 inference-time `<pause>` tokens. The design of *zero-delay-robust* pause-trained models is thus an important question for future work.

APPENDING VS. PREPENDING PAUSES    In our main experiments, we chose to append `<pause>` tokens since it is the most natural format for a general setting e.g., in long-text-generation as in a conversational agent, one would append `<pause>` tokens to the current text rather than deploying the tokens all at once at the beginning of the conversation. Furthermore, when there is unidirectional attention, prepending these tokens should make no difference. Nevertheless, in our downstream tasks which use bidirectional attention on the prefix, it makes sense to consider prepending `<pause>` tokens. We investigate this in Table 2 in Appendix C. Most importantly, we find that, for `PausePT_PauseFT`, even prepending the `<pause>` token performs improves over standard end-to-end training. However, appending is still the more optimal choice. This indicates that pause-pretraining induces considerable biases in how readily the delays are used based on their positional embeddings.

## 6   DISCUSSION AND KEY OPEN QUESTIONS

**Enhanced computational width.** One hypothesis as to why Transformer delays can help is that it increases expressivity by increasing the computational *width*. To produce the $(K+1)^{\text{th}}$ token, standard inference involves a computational depth of $L$ (corresponding to the *sequential* computation of $L$ layers), and a computational width of $K$ (corresponding to the *parallel* $K$ computations per layer). With $M$ `<pause>` tokens however, we perform $K+M$ parallel computations. We hypothesize that this additional width helps certain downstream tasks. Take for example, comprehension-based question-answering tasks. Here, having a greater number of attention units per layer, would permit a finer distribution of attention across various parts of the supporting context (where the answer resides). We speculate that this would allow the lower layers to extract more precise and diverse representations, which a higher layer can more reliably aggregate to produce a final answer.

We formalize this in Theorem J.1 (stated informally below). Our key theoretical insight is that the attention module can have a high "raw" parameter-count-based capacity, but low "implementation capacity": the number of operations implemented for a given input, which is bottlenecked by the number of tokens. Pause tokens can help relieve this bottleneck and tap into the raw representational capacity. We hope this preliminary result can inspire further discussions on how to formalize the Transformer's implementation capacity and differentiate it from the raw parameter count.

**Theorem 6.1.** *(informal; see Theorem J.1) Assume that the attention module has sufficiently many parameters $(K)$ that it is much larger than the number of inputs tokens $(L)$. Then there are tasks that involve many independent computations $N$, where $N > L$ (but $N < K$), such that a 2-layer Transformer can implement the task if and only if it uses pause tokens.*

**Computational expansion along with parameter expansion.** How do the gains offered by `<pause>` tokens vary with parameter count? An immediate hypothesis would be that for smaller models, delays become more beneficial as they provide a much-needed capacity increase in an otherwise capacity-deprived model. But a preliminary comparison between our two model sizes surprisingly suggests the opposite. Intuitively, we conjecture that a smaller model may simply not have enough raw capacity to implement a variety of distinct computations to utilize the new pathways introduced by `<pause>` tokens. This intuition is also echoed in Theorem 6.1 where smaller models would break our assumption that parameter count $K$ is large enough.

**Computational expansion vs parameter expansion.** There are trivial ways to extend the next-token computation process, say via more attention heads or more layers. For a fixed inference-time FLOPS budget, can these give similar gains as pause tokens? In Appendix I, we argue why this is not the case — attaining similar gains requires significant parameter expansion and a significant expansion of the FLOPS budget. Thus, it is both practically and theoretically remarkable that pause tokens can yield gains despite negligible addition to the parameter count.

**Pause-inference vs. Chain-of-Thought.** It is worth contrasting the above computational advantage with that enjoyed by chain-of-thought (CoT) prompting (Wei et al., 2022). Here, one prompts the model to generate a long series of intermediate reasoning steps before producing the final answer. Thus, CoT also corresponds to greater computational width, by way of delaying its final answer

(albeit with meaningful tokens). However, CoT has a vital added advantage: it also increases the computational depth to a significant extent (Feng et al., 2023, Theorem 3.3 and 3.4). In particular, each (meaningful) delay token generated by CoT is autoregressively generated by the model. Thus, if there are $M$ such tokens and $L$ layers, the final token arises out of roughly $M \cdot L$ sequentially composed operations. Thus, CoT has a computational depth that is larger by a *multiplicative* factor of $M$, compared to pause-inference. Finally, we note that one major advantange of CoT is that it does not seem to require any special modifications in the pretraining objective.

## 7    RELATED WORK

MEMORY TOKENS    Multiple works have proposed using dummy tokens as a ways of introducing memory into the model. Closest to our work is Burtsev et al. (2020) who *prepend* these tokens (rather than append them) and crucially, introduce them only during training and inference on the target tasks. On smaller scratch-trained models (with parameter counts of 10M, 65M and 277M) and a pretrained BERT model (109M), this reportedly gives minimal gains. This echoes our own mixed results for the comparable `StdPT_PauseFT` variant, and the fact that our smaller model shows gains on fewer datasets. Orthogonally, Sukhbaatar et al. (2019) introduce a large set of "global" memory tokens (of the order of ten million new parameters) across all layers. However, these are independent of the input, and only act as keys and values (not as queries). For vision transformers (ViTs), AdaTape (Xue et al., 2023) appends tokens from a learnt memory bank. Concurrent work in ViTs (Darcet et al., 2023), use multiple different "register tokens" appended to the image patches. In the context of recurrent networks, closest to us is Herel & Mikolov (2023) who train with "thinking" tokens for extra compute for the target task and find small perplexity gains on reasoning tasks. Grave et al. (2016) use a memory cache as a way to attend to past hidden vectors.

**Adaptive Compute Transformers**    *Adaptive compute* methods provide users with flexibility in inference time depending on the input to the model. Although adaptive compute is not the objective of this paper (after all, we use the same number of pause tokens across all inputs), pause-inference can be viewed as a potential basis for a future adaptive compute method. The Universal Transformer (Dehghani et al., 2019) adaptively increases *serial* computations by repeating certain layers. Graves (2017) explored a similar recurrence over hidden units in the context of RNNs. A future adaptive-compute version of pause-inference would fall in a much less explored paradigm that varies the number of input tokens, akin to AdaTape in ViTs (Xue et al., 2023).

We discuss other relevant works in detail in Appendix K.

## 8    CONCLUSION, LIMITATIONS AND FUTURE WORK

Pause-training takes a step beyond the paradigm of "immediate" next-token prediction in language models. The key idea is to train models with (dummy) `<pause>` tokens so that the model can learn to harness the additional inference-time computation. This can improve performance on a variety of tasks, *if* we train with `<pause>` tokens both during pretraining and downstream finetuning.

However, by extension of the fact that every downstream task has an optimal number of `<pause>` tokens, we do not claim that pause-training should benefit every downstream task. Some tasks may simply be better off with zero `<pause>` tokens. The most important limitation though is that the expense of pause-pretraining comes in the way of making this idea more widely accessible. Consequently, we do not study how the gains generalize across more model sizes (beyond 1B and 130M), or to encoder-decoder architectures, or to other pretraining mixtures and objectives. Next, while we have laid out some preliminary theory for why `<pause>` tokens may be beneficial, we leave a rigorous understanding for future study. We also leave open a variety of follow-up algorithmic questions: pause-training with multiple *different* `<pause>` tokens, better determining the number of `<pause>` tokens (perhaps using model confidence), inducing robustness to shifts in delays, and so on. But the most pressing next step would be to find ways to make delays helpful directly on a standard pretrained model. Overall, we hope that our work opens up many avenues for theoretical and practical work in the paradigm of delayed next-token prediction.

**Acknowledgements**: We would like to thank Kaifeng Lyu, Srinadh Bhojanapalli, Yongchao Zhou, Nikunj Saunshi, and Seungyeon Kim for helping set up the initial codebase and for their guidance.

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

## A    PRELIMINARIES: TRANSFORMER

Consider a vocabulary $\mathcal{V}$ and an input $\mathbf{p}_{1:K} \in \mathcal{V}^K$ of $K$ tokens, and an $L$-layer decoder-only language model. The $l$'th layer of the Transformer produces one intermediate vector for each token here, denoted by $\mathbf{v}_k^{(l)} \in \mathbb{R}^D$ for $k = 1, \ldots, K$. We first describe this operation before outlining the end-to-end next-token generation process.

Consider a Transformer (Vaswani et al., 2017) block $T(\cdot) : \mathbb{R}^{K \times D} \to \mathbb{R}^{K \times D}$ that operates over a sequence of $K$ intermediate vectors. The block is defined by $H$ many sets of four matrices each, $W_{\text{query}}^{(h)}, W_{\text{key}}^{(h)}, W_{\text{value}}^{(h)}$ and $W_{\text{out}}^{(h)}, \in \mathbb{R}^{D_{\text{attn}} \times D}$ (for $h = 1, \ldots H$ each denoting an attention head), and a single parameterized feedforward module $f_{\text{FF}} : \mathbb{R}^D \to \mathbb{R}^D$. Let $\Phi_{\text{LN}} : \mathbb{R}^D \to \mathbb{R}^D$ denote the layer-norm operation. Given the input vectors $V_{1:K} \in \mathbb{R}^{D \times K}$, the output $V'_{1:K}$ of the Transformer block $T(\cdot)$ can be expressed in the following steps. For all $k \leq K$,

$$\mathbf{a}_k = \Phi_{\text{LN}}^{(1)} \left( \mathbf{v}_k + \sum_{h=1}^{H} (W_{\text{out}}^{(h)})^T \cdot W_{\text{value}}^{(h)} V_{1:k} \text{softmax} \left( \frac{(W_{\text{key}}^{(h)} V_{1:k})^T W_{\text{query}}^{(h)} \mathbf{v}_k}{\sqrt{D_{\text{attn}}}} \right) \right) \qquad (4)$$

$$\mathbf{v}'_k = \Phi_{\text{LN}}^{(2)} \left( \Phi_{\text{FF}}(\mathbf{a}_k) + \mathbf{a}_k \right). \qquad (5)$$

Here, the first step computes $K$ different self-attention outputs by attending to all $K$ input vectors, while the second step individually processes each attention output via a feedforward network and other normalization components to produce the final output of the block. Note that here we have assumed a unidirectional attention mechanism; for a bidirectional mechanism over the whole $K$-length prefix, one simply needs to replace $V_{1:k}$ with $V_{1:K}$ in the above computation.

Given this block, the Transformer generates the next token as follows. Let $\Phi_{\text{token}} : \mathcal{V} \to \mathbb{R}^D$ and $\Phi_{\text{position}} : \mathbb{N} \to \mathbb{R}^D$ denote the token-embedding and position-embedding layers. With an abuse of notation, let the token unembedding layer be denoted as $\Phi_{\text{token}}^{-1}$, which maps from $\mathbb{R}^D$ to a probability vector in $\Delta^{|\mathcal{V}|-1}$. Let $T^{(l)}(\cdot)$ denote the $l^{\text{th}}$ Transformer layer. Then, the Transformer commits the following operations in sequence to arrive at the $(K + 1)^{\text{th}}$ token.

$$\mathbf{v}_k^{(0)} = \Phi_{\text{token}}(p_k) + \Phi_{\text{position}}(k) \qquad (6)$$

$$V_{1:K}^{(l)} = T^{(l)}(V_{1:K}^{(l-1)}), \forall l \in [1, L] \qquad (7)$$

$$p_{K+1} \sim \Phi_{\text{token}}^{-1}(\mathbf{v}_K^{(L)}). \qquad (8)$$

For a more detailed mathematical exposition of the Transformer model, we refer the reader to Thickstun (2021).

## B    ADDITIONAL DOWNSTREAM FINETUNING RESULTS

We first report the downstream finetuning performance for the 1B model in Table 1 (numbers corresponding to Figure 3 in §4.3). Further, in Figure 5 we report downstream performance on various tasks for a 130M decoder-only model. Again we observe that PausePT_PauseFT clearly outperforms standard training baseline (StdPT_StdFT) on GSM8k, CommonSenseQA, LAMBADA and on our fact recall tasks like WebQA and NaturalQA. However, surprisingly, we do not observe gains on SQuAD, in contrast to the gains observed in 1B model. Overall, we see an improvement in six tasks for the smaller model (one of which is PhysicalIQA where the gain is minimal).

## C    PREPENDING VS APPENDING PAUSE TOKENS

In Section 5, we discussed the effect of prepending the pause token in comparison to the default approach of appending them to the end of prefix. Table 2 compares the two approaches. As stated before in Section 5, for the PausePT_PauseFT training algorithm, we observe that prepending the pause tokens still outperforms the baseline but is (slightly) worse than appending the pause tokens on some benchmarks like GSM8k and SQuAD. For StdPT_PauseFT however, we see mixed results with equal number of wins and losses between the prepending and appending.

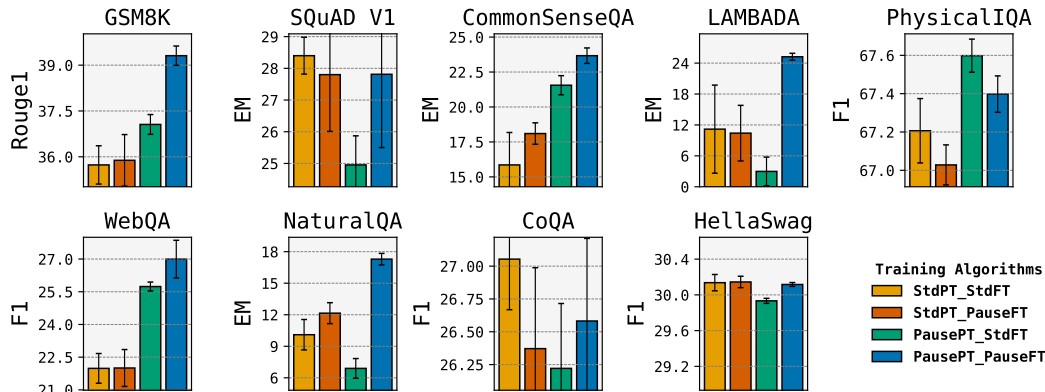

Figure 5: **Downstream performance of pause-training on a 130M decoder-only model.** We find on six out of our nine downstream tasks, the pause-pretrained and pause-finetuned model (`PausePT_PauseFT`) outperforms standard training (`StdPT_StdFT`) on a 130M decoder-only model. For example, on the reasoning task of GSM8k, we observe $3\%$ gains in Rouge1 scores (we compare Rouge1 as the final accuracy was too low to be meaningful for our 130M model). Similarly on the general understanding task of CommonSenseQA, we observe upto $8\%$ gains. We note here that we solve CommonsenseQA as a decoding task and not rank classification task, and hence report the Exact Match scores. We also highlight that while pause-trainined on the 1B model showed significant gains on SQuAD, they disappear here.

| Dataset | Metric | StdPT_StdFT | StdPT_PauseFT | | PausePT_StdFT | PausePT_PauseFT | |
|---|---|---|---|---|---|---|---|
| | | | 10 | 50 | | 10 | 50 |
| GSM8k | Acc | $7.5 \pm 0.5$ | $6.9 \pm 1.0$ | $6.5 \pm 0.8$ | $7.7 \pm 0.5$ | $8.5 \pm 0.9$ | $7.7 \pm 0.3$ |
| | Rouge1 | $42.3 \pm 0.5$ | $41.7 \pm 0.7$ | $41.2 \pm 1.3$ | $43.5 \pm 0.1$ | $44.2 \pm 0.2$ | $44.1 \pm 0.2$ |
| SQuAD | EM | $36.4 \pm 2.5$ | $36.6 \pm 2.2$ | $40.2 \pm 3.2$ | $38.4 \pm 2.9$ | $51.7 \pm 2.3$ | $55.9 \pm 1.0$ |
| CommonSense QA | EM | $26.9 \pm 2.9$ | $28.8 \pm 2.8$ | $28.7 \pm 2.0$ | $27.7 \pm 2.7$ | $34.8 \pm 1.2$ | $32.3 \pm 0.8$ |
| LAMBADA | EM | $16.4 \pm 1.7$ | $18.4 \pm 0.3$ | $18.5 \pm 0.6$ | $13.7 \pm 5.1$ | $18.8 \pm 0.1$ | $18.5 \pm 0.2$ |
| Web Questions | EM | $13.7 \pm 2.1$ | $9.0 \pm 4.4$ | $12.4 \pm 2.6$ | $15.0 \pm 2.5$ | $13.8 \pm 3.7$ | $16.0 \pm 1.6$ |
| Natural Questions | EM | $23.6 \pm 1.2$ | $24.3 \pm 1.4$ | $23.9 \pm 1.3$ | $24.3 \pm 7.5$ | $24.9 \pm 1.3$ | $26.9 \pm 0.4$ |
| CoQA | F1 | $29.9 \pm 1.0$ | $30.7 \pm 0.5$ | $30.3 \pm 0.5$ | $31.1 \pm 0.3$ | $31.3 \pm 1.1$ | $31.6 \pm 0.5$ |
| PhysicalIQA | F1 | $73.3 \pm 0.2$ | $73.9 \pm 0.2$ | $74.0 \pm 0.2$ | $74.1 \pm 0.2$ | $74.1 \pm 0.1$ | $74.2 \pm 0.2$ |
| HellaSwag | F1 | $37.8 \pm 0.1$ | $37.9 \pm 0.2$ | $37.8 \pm 0.2$ | $37.9 \pm 0.1$ | $37.7 \pm 0.2$ | $37.8 \pm 0.2$ |

Table 1: **Downstream performance on various tasks for the 1B decoder-only model.** We observe that `PausePT_PauseFT` outperforms the standard training baseline on 8 out of the 9 tasks considered in this work. See §4.3 and Figure 3 for further details.

# D    ZERO-SHOT EVALUATION

In §4.3, we showed the efficacy of pause-pretrained models when finetuned with pause tokens. However, we note that we witness gains even in the zero-shot setting, where the model is not finetuned. In Figure 6, we compare the zero-shot accuracy of the standard pretrained 1B model with that of pause-pretrained version. For the pause-pretrained model, we perform evaluation with 0, 10 and 50 `<pause>` tokens appended to the prefix. Observe that pause-pretraining gives some gains on tasks like GSM8k and HellSwag. However, we note here the absolute value of our zero-shot accuracies are quite low, as we experiment with a small 1B parameter model.

| Dataset | Metric | StdPT_StdFT | StdPT_PauseFT | | PausePT_PauseFT | |
|---------|--------|-------------|---------------|---|-----------------|---|
| | | | Prepending | Appending | Prepending | Appending |
| **GSM8k** | Acc. | $7.5 \pm 0.5$ | $8.0 \pm 1.0$ | $6.9 \pm 1.0$ | $8.0 \pm 0.4$ | $8.5 \pm 0.9$ |
| **SQuAD** | EM | $36.4 \pm 2.5$ | $35.0 \pm 1.5$ | $40.2 \pm 3.2$ | $44.0 \pm 3.2$ | $55.9 \pm 1.0$ |
| **CommonQA** | EM | $26.9 \pm 2.9$ | $31.0 \pm 1.3$ | $28.8 \pm 1.5$ | $34.5 \pm 1.0$ | $34.8 \pm 1.2$ |
| **Lambada** | EM | $16.4 \pm 1.7$ | $17.8 \pm 0.4$ | $18.5 \pm 0.6$ | $18.0 \pm 1.1$ | $18.8 \pm 0.1$ |
| **PhysicalIQA** | F1 | $73.3 \pm 0.2$ | $74.0 \pm 0.3$ | $74.0 \pm 0.3$ | $74.2 \pm 0.2$ | $74.2 \pm 0.2$ |
| **NaturalQ** | EM | $23.6 \pm 1.2$ | $24.1 \pm 0.6$ | $24.3 \pm 1.4$ | $25.7 \pm 0.9$ | $26.9 \pm 0.4$ |

Table 2: **Prepending vs. appending the pause tokens** (§5). We observe that prepending the pause tokens still outperforms the standard training baseline of StdPT_StdFT, but is suboptimal to appending the <pause> tokens for PausePT_PauseFT training algorithm. However, for StdPT_PauseFT, both have equal number wins and losses.

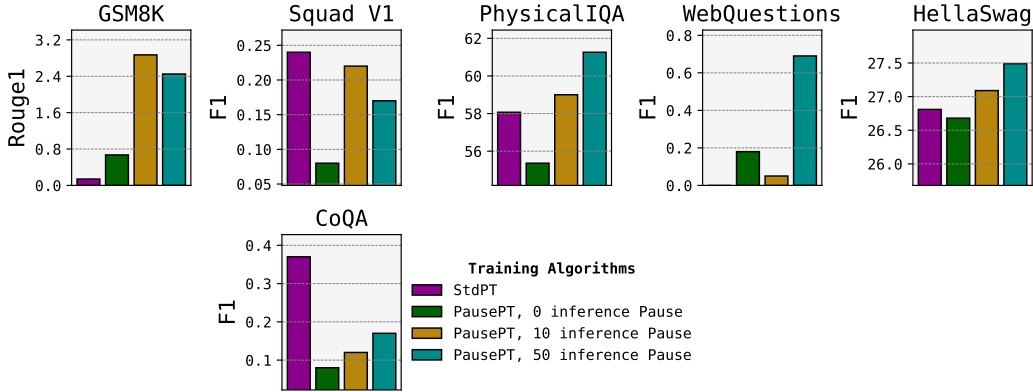

Figure 6: **Zero-shot evaluation of pause-pretrained models.** Zero-shot inference with pause tokens on a pause-pretrained model gives gains on tasks like GSM8k and HellaSwag. However, we note that our zero-shot accuracies are quite low, as we experiment with a small 1B parameter model.

# E    VARYING NUMBER OF PAUSE TOKENS $M_{ft}$

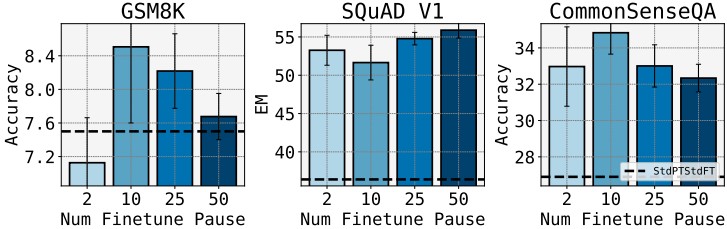

Figure 7: **Varying finetuning delay**: We examine the effect of varying the number <pause> tokens used in downstream finetuning ($M_{\text{ft}}$, §5) on the performance. Typically, we observe that there exists an optimal number of <pause> tokens as expected for each dataset.

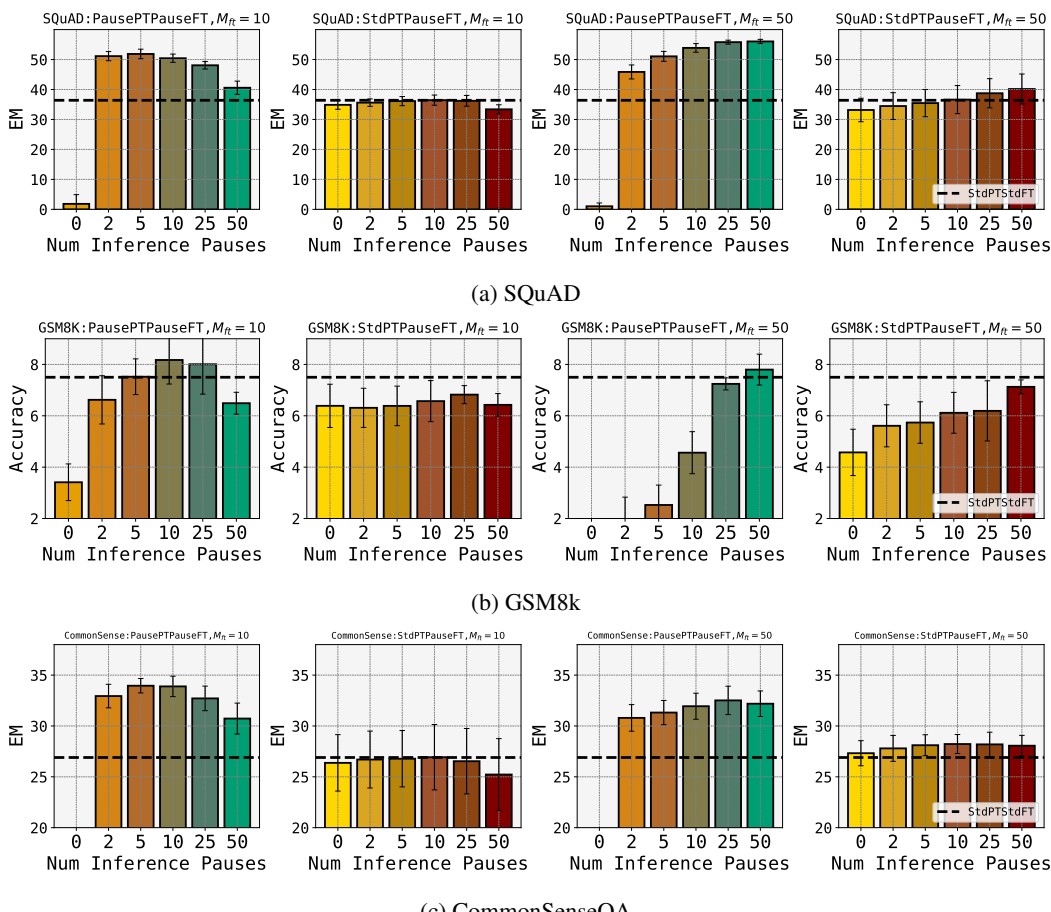

(a) SQuAD

(b) GSM8k

(c) CommonSenseQA

Figure 8: **Varying inference-time delays:** We test the robustness of pause-trained models to varying number of inference time <pause> tokens (setting $M_{\mathrm{inf}}$ not equal to $M_{\mathrm{ft}}$), which exposes the model to a serious test-time distribution shift (§5). Pause-training degrades gracefully to shifts as wide as $M_{\mathrm{inf}} \in [5, 25]$ for $M_{\mathrm{ft}} = 10$ and $M_{\mathrm{ft}} = 50$ both for `PausePT_PauseFT` and `StdPT_PauseFT`, apart from GSM8k wherein there is a drop for $M_{\mathrm{ft}} = 50$. In each row, the first and the third column considers the `PausePT_PauseFT` model for $M_{\mathrm{ft}}$ set to 10 and 50, respectively. Likewise, the second and the fourth column show the same for `StdPT_PauseFT` model.

In Figure 7, we study the effect of varying the number of pause tokens used during downstream finetuning ($M_{ft}$) on the downstream performance. We refer the reader to §5 for further details. Again we observe that there exists an optimal number of pause tokens to be used during downstream finetuning, depending on the task.

# F  ROBUSTNESS TO VARYING NUMBER OF INFERENCE TIME PAUSES

Recall in §5 and Figure 4c we observed that pause-training is robust to using a different number of inference time pauses compared to that used during finetuning (i.e. $M_{\mathrm{inf}} \neq M_{\mathrm{ft}}$). We present additional results regarding the same in Figure 8a, Figure 8b and Figure 8c. Again, we observe that the performance degrades gracefully for the pause-trained models, even with shifts that halve the number of tokens seen. However, we still find a drastic drop in performance when no delay is given during inference for the `PausePT_PauseFT` model.

| Dataset | Learning Rate | Warmup Steps | Finetuning Steps | Batch Size |
|---|---|---|---|---|
| **SQuAD** | 1.0E-04 | 100 | 10000 | 256 |
| **GSM8k** | 1.0E-04 | 200 | 20000 | 16 |
| **HellaSwag** | 5.0E-06 | 100 | 1000 | 16 |
| **PhysicalIQA** | 1.0E-06 | 50 | 600 | 32 |
| **CoQA** | 5.0E-05 | 75 | 3500 | 16 |
| **CommonSenseQA** | 5.0E-05 | 100 | 4000 | 16 |
| **LAMBADA** | 5.0E-05 | 40 | 2800 | 16 |
| **WebQuestions** | 5.0E-04 | 200 | 2000 | 16 |
| **NaturalQuestions** | 1.0E-04 | 100 | 5000 | 256 |

Table 3: Downstream finetuning hyperparameters for the 1B model.

## G  DOWNSTREAM DATASET DESCRIPTION

We finetune and evaluate the pretrained models (both standard and pause-pretrained) on the following datasets:

1. GSM8k: A reasoning task with 8.5k grade school math word problems (Cobbe et al., 2021).

2. SQuAD V1: Reading-comprehension task based on Wikipedia (Rajpurkar et al., 2016).

3. CommonSenseQA: Requires different types of commonsense knowledge to choose the correct answer (Talmor et al., 2019). Our implementation of CommonSenseQA is as a decoding task, and hence we report Exact Match (EM) scores.

4. LAMBADA: Text-understanding task requiring last-word prediction based on a long context (Paperno et al., 2016).

5. Web Questions: A fact-recall dataset of commonly-asked questions on the web (Berant et al., 2013).

6. PhysicalIQA: A physical commonsense reasoning dataset, which test the ability to understand interactions with the world (Bisk et al., 2020).

7. Natural Questions: QA task which requires answering fact-based questions from Wikipedia article pages (Kwiatkowski et al., 2019). Since we use the closed-book version of this dataset (no access to helpful context), this is a fact-recall task.

8. HellaSwag: Next-sentence prediction task based on common-sense inference (Zellers et al., 2019).

9. CoQA: Question-answering task based on a context (Reddy et al., 2019).

## H  HYPERPARAMETERS: DOWNSTREAM FINETUNING

We share all the hyperparameters for downstream finetuning in Table 3 (1B model) and Table 4 (130M model). We also provide the decoder-only architecture details for the two models considered in this work in Table 5.

| Dataset | Learning Rate | Warmup Steps | Finetuning Steps | Batch Size |
|---------|---------------|--------------|------------------|------------|
| **SQuAD** | 1.00E-04 | 400 | 40000 | 16 |
| **GSM8k** | 1.00E-04 | 75 | 7500 | 16 |
| **CommonSenseQA** | 5.00E-05 | 100 | 6000 | 16 |
| **LAMBADA** | 5.00E-05 | 40 | 1400 | 16 |
| **WebQuestions** | 5.00E-04 | 200 | 2000 | 16 |
| **NaturalQuestions** | 5.00E-04 | 100 | 5000 | 256 |
| **CoQA** | 1.00E-04 | 75 | 3500 | 16 |
| **PhysicalIQA** | 1.00E-06 | 50 | 600 | 32 |
| **HellaSwag** | 1.00E-06 | 100 | 1000 | 16 |

Table 4: Downstream finetuning hyperparameters for the 130M model.

| Model | 130M | 1B |
|-------|------|-----|
| **Parameters** | 136,237,056 | 1,345,085,440 |
| **Transformer Layers** | 12 | 24 |
| **Attention Heads** | 12 | 32 |
| **Embedding Dimension** | 768 | 2048 |
| **Hidden Dimension** | 3072 | 8092 |

Table 5: Architecture details for the models considered in this work

---

**Algorithm 1:** Pause-pretraining

---

**Pretraining with Pause**

**Inputs:** Pretraining dataset $\mathcal{D}_{\mathrm{pt}}$, decoder-only model $f_\theta$, number of <pause> tokens $M_{\mathrm{pt}}$ to insert, pause token <pause>

$\mathbf{p}_{1:N} \sim \mathcal{D}_{\mathrm{pt}}$  /* Input Sequence from corpus */

$\tilde{\mathbf{p}}_{1:N+M_{pt}} = \mathsf{random\_insert}(\mathbf{p}_{1:N}, \texttt{<pause>}, M_{\mathrm{pt}})$  /* Insert $M_{pt}$ pause tokens randomly in the original input sequence $\mathbf{p}_{1:N}$, extending its length by $M_{pt}$ */

$S_{\mathrm{ignore}} = \{k \in [0, N + M_{pt} - 1] : \tilde{p}_{k+1} = \texttt{<pause>}\}$  /* Identify the set of positions where the next token is <pause> */

$\mathcal{L}_{\mathrm{PausePT}}(f_\theta, \tilde{\mathbf{p}}_{1:N+M_{pt}}) = \sum_{k=1, k \notin S_{\mathrm{ignore}}}^{N+M_{pt}-1} \mathcal{L}_{\mathrm{CE}}(\tilde{p}_{k+1}, f_\theta(\tilde{\mathbf{p}}_{1:k}))$  /* Next token prediction error excludes targets which are pause (model isn't made to learn to predict pause itself) */

$\theta = \theta - \nabla_\theta \mathcal{L}_{\mathrm{PausePT}}(f_\theta, \tilde{\mathbf{p}}_{1:N+M_{\mathrm{pt}}})$  /* Update the model */

---

---

**Algorithm 2:** Pause-finetuning

---

**Stage 2: Finetuning with Pause**

    **Inputs:** Downstream labeled dataset $\mathcal{D}_{\text{ft}}$, pretrained model $f_\theta$, number of `<pause>` tokens $M_{\text{ft}}$ to insert, pause token `<pause>`

    $\mathbf{p}_{1:N}, \mathbf{t}_{1:T} \sim \mathcal{D}_{\text{ft}}$                                    `/* Sample prefix and target */`

    $\tilde{\mathbf{p}}_{1:N+M_{\text{ft}}} = \text{Concat}[\mathbf{p}_{1:N}, [\texttt{<pause>}] \times M_{\text{ft}}]$              `/* Append prefix and` $M_{\text{ft}}$ `pauses */`

    $\mathcal{L}_{\text{PauseFT}}(f_\theta, \tilde{\mathbf{p}}_{1:N+M_{\text{ft}}}, \mathbf{t}_{1:T}) = \sum_{k=0}^{T-1} \mathcal{L}_{\text{CE}}(t_{k+1}, f_\theta(\text{Concat}[\tilde{\mathbf{p}}_{1:N+M_{\text{ft}}}, \mathbf{t}_{1:k}]))$    `/* Next token prediction error on targets */`

    $\theta = \theta - \nabla_\theta \mathcal{L}_{\text{PauseFT}}(f_\theta, \tilde{\mathbf{p}}_{1:N+M_{\text{ft}}}, \mathbf{t}_{1:T})$

---

**Algorithm 3:** Pause-inference

---

**Stage 3: Inference with Pause**

    **Inputs:** Prefix $\mathbf{p}_{1:N}$, finetuned model $f_\theta$, number of `<pause>` tokens $M_{\text{inf}}$ to insert, Pause token `<pause>`

    $\tilde{\mathbf{p}}_{[1:N+M_{\text{inf}}]} = [\mathbf{p}_{1:N}, [\texttt{<pause>}] \times M_{\text{inf}}]$              `/* Append` $M_{\text{inf}}$ `pauses to prefix */`

    $\tilde{\mathbf{p}}_{N+M_{\text{inf}}+1} \sim f_\theta(\tilde{\mathbf{p}}_{1:N+M_{\text{inf}}})$        `/* Predict the next token in the sequence, and continue in an auto-regressive fashion */`

---

# I    INFERENCE TIME COST OF PAUSE TOKENS

One way to assess the inference-time compute efficiency of a method is by estimating the number of *Floating Point Operations per Second (FLOPS)* it requires. A related, but independent metric, is the *Wall Clock Time* as it affects the latency of deployed systems. Below, we analyze how efficient pause-inference is along these two metrics in this section. Broadly, we make two arguments:

1. Pause-inference offers a more FLOPS-efficient way of increasing performance, as compared to other natural ways of expanding the number of attention operations in a Transformer, such as by adding layers or by adding attention heads.

2. Pause-inference is also wall-clock-efficient compared to the above techniques as it virtually introduces no overhead. When compared to CoT, pause-inference provides a computationally more granular and cheaper way to improve performance (although still upper-bounded by CoT in terms of *performance*).

## I.1    PAUSE TOKENS ALLOW FOR A MORE EFFICIENT USE OF FLOPS

We frame our FLOPS-efficiency analysis as follows. Consider introducing $p$ pause tokens during inference in a given Transformer. How many additional FLOPS does this require? If we spent the same budget of additional FLOPS to introduce more attention operations via other techniques — namely, via an appropriate number of additional layers or additional attention heads — would we find a similar improvement in quality, in terms of metrics like accuracy?

Concretely, we use a running example of the 1B model used in this paper, whose architecture details are provided in Table 5. Specifically, we have number of transformer layers as $l = 24$, input embedding dimension as $h = 2048$, per attention head embedding dimension $d = h/a$, where $a = 32$ is the number of attention heads. We also anchor our analysis for input prompts with $n = 100$ tokens, which represents the average prompt length for many tasks considered in this work. For our analysis, we rely on supporting lemmas deferred to Section I.2.1.

**FLOPS-efficiency of $p$ pause tokens vs $k$ additional layers:**    In the context of the downstream task of SQuAD, appending $p = 10$ pause tokens yields an $18\%$ increase in EM score. Applying these values ($n = 100, p = 10$) and $l = 24$ in the FLOPS-efficiency analysis of Theorem I.3, we can deduce that if we were to allocate the same FLOPS budget to adding more layers to the Transformer stack, we can at most add 2 layers. This enhancement corresponds to a modest $10\%$ rise in parameter count, expanding the model from a 1B parameter model to a 1.1B parameter model. However, in practice, when scaling the parameter count, significant performance improvements (such an $18\%$ increase in EM score) are typically observed only when the parameter count is scaled by much larger factors. Thus we argue that in this case, pause tokens provide a more inference-time-FLOPS-efficient way of increasing performance. Conceptually, this underscores the fact that pause tokens introduce an alternative dimension to representation capacity, distinct from the traditional approach of scaling the parameter count.

**Comparing FLOPS with increase in attention heads:**    In the standard Transformer implementation used in practice, when one increases the attention heads ($a$), although the number of attention operations increases, the per-attention-head embedding dimension ($d$) proportionally gets reduced to keep the overall embedding dimension constant ($h$). Thus, there is effectively no change in the number of FLOPS. In contrast, adding pause tokens increases the number of attention operations, while keeping the per-attention-head embedding dimension $d$ constant. Therefore, for a fair comparison, we consider the case where we increase $a$, while keeping the per-attention-head dimension $d$ fixed (and we fix it to be equal to $h$).

Then, from our analysis in Theorem I.4 we have that for an input of length $n = 100$ and $a = 32$ attention heads, appending $p = 10$ pause tokens is equivalent to increasing the number of attention heads by $k = 3$. However, increase the attention heads by 3 in the 1B model configuration adds only 48M parameters (we add $W_q, W_k, W_v, W_{proj} \in \mathbb{R}^{h \times h}$ per additional attention head), bringing the model to a parameter count of 1.048B from 1B. This, we argue cannot account for any significant performance improvement equivalent to the improvements seen under pause-training proposed in this paper.

## I.2 PAUSE TOKENS DO NOT ADD SEQUENTIAL COMPUTE

**Comparing pause tokens with adding layers or attention heads**  Recall that `<pause>` tokens are added as a part of the input prompt, where each token is processed in parallel. Thus, `<pause>` tokens do not add extra serial computations. If there are sufficiently many parallel threads available, the wall clock overhead from pause tokens would be a minimal percentage of the time required for standard inference. However, in contrast, increasing the number of transformer layers increases the length of sequential operations, causing the wall-clock time to increase proportionally to the fraction of layers added. Note that adding attention heads should have a similar effect as adding pause tokens, as they too introduce parallel, not sequential operations.

**Comparison with Chain-of-Thought (CoT) prompting**  Recall that CoT involves autoregressively decoding a long sequence of tokens involved in the model's reasoning. This requires a significant wall clock time cost, proportional to $O(pl)$, if $p$ is the number of intermediate reasoning tokens and $l$ is the number of layers. In stark contrast, pause tokens do not add extra wall-clock time. Furthermore, CoT prompting offers little flexibility in how large $p$ can be. Pause-inference on the other hand, offers a more direct way for manipulating the number of pause-tokens (even if, in its current version, this adaptivity is not robust beyond a point).

### I.2.1 SUPPORTING LEMMAS FOR ESTIMATING FLOPS EFFICIENCY

In Lemma I.1, we present the facts about FLOPS required for basic vector and matrix calculations. Subsequently, in Lemma I.2, we compute the overall FLOPS required for an end-to-end Transformer computation. Finally, in Theorem I.3 and I.4, we derive how different kinds of parameter expansions in the model compare to adding pause tokens, in terms of FLOPS efficiency. Specifically, Theorem I.3 establishes the number of layers one needs to add to a model to realize the same number of FLOPS as adding $p$ pause tokens. Theorem I.4 presents a similar result for adding attention heads.

**Lemma I.1.** *(FLOPS for vector and matrix calculations) The number of flops required to compute:*

1. *the dot product $\mathbf{v}_1 \cdot \mathbf{v}_2$ where $\mathbf{v}_1, \mathbf{v}_2 \in \mathbb{R}^d$ is $O(d)$.*

2. *the matrix multiplication $M_1 \cdot M_2$, where $M_1 \in \mathbb{R}^{a \times b}$ and $M_2 \in \mathbb{R}^{b \times c}$, the FLOPS is $O(a \times b \times c)$.*

*Proof.* For a dot product between $\mathbf{v}_1$ and $\mathbf{v}_2$, both of dimension $d$, the total number of FLOPS is given by the sum of multiplications and additions required. Specifically, it involves $d$ multiplications and $d-1$ additions, totaling to $2d-1$ FLOPS. For the ease of calculation, we approximate this as $2d$.

For a matrix multiplication of $M_1 \in \mathbb{R}^{a \times b}$ and $M_2 \in \mathbb{R}^{b \times c}$, each element of the resulting matrix is computed by taking the dot product of a row from $M_1$ and a column from $M_2$, which requires $2b$ FLOPS . Since there are $a \times c$ such dot products to compute for the entire matrix multiplication, the total FLOPS amount to $(2b) \times a \times c$. However, for simplicity, if we only consider the multiplicative operations, the FLOPS reduce to $a \times b \times c$. $\square$

**Lemma I.2.** *(FLOPS for one end-to-end Transformer computation) Consider an $l$ layered decoder only language model, where we denote input embedding dimension as $h$, number of attention heads as $a$ and per-attention-head embedding dimension as $d$. We assume feed-forward hidden dimension to be $4h$ and finally let $n$ denote the length of input sequence. Then the total FLOPS are given as:*

$$F(n, h, d, a, l) = (4nadh + 2an^2d + 8nh^2)l \tag{9}$$

We note that standard Transformer implementations assume $d = h/a$, i.e. the per-attention-head embedding dimension decreases as the number of attention heads are increased. However, we treat these are three independent hyperparameters for greater flexibility in our analysis.

*Proof.* Let us consider the various per-layer operations in a decoder-only model step-by-step and count their FLOPS :

1. **q**, **k**, **v** vector computation: Given input token $x \in \mathbb{R}^h$, for the query vector computation, we have $Q^j = W_q^j x \forall j \in [1, a]$, where $W_q^j \in \mathbb{R}^{d \times h}$. The same extends for key and value vector computations. Thus total FLOPS required is $3nadh$.

2. Self-attention: Given $Q \in \mathbb{R}^{n \times d}$ and $K \in \mathbb{R}^{n \times d}$, $QK^T$ incurs $n^2 d$ flops. The obtained $\alpha = softmax(\frac{QK^T}{\sqrt{d}}) \in \mathbb{R}^{n \times n}$ is multiplied by $V \in \mathbb{R}^{n \times d}$, costing another $n^2 d$ flops. Note that for simplicity, we ignore that FLOPS from softmax or the division by $\sqrt{d}$ operation as they are negligible. Thus, the total FLOPS = $a[n^2 d + n^2 d] = 2an^2 d$.

3. Combining multi-head-attention: The MHA projection matrix concatenates all the outputs from individual attention heads above and projects them to output of dimension $h$. For simplicity, we ignore the FLOPS from the skip connection as it adds only a relatively minimal number. Thus total FLOPS = $n \times h \times ad = nadh$.

4. Feed-forward network: This adds another $8nh^2$ FLOPS . Again, for simplicity we ignore the FLOPS from the skip connection.

Combining the FLOPS from each of the sub-parts above, we have:

$$FLOPS = (4nadh + 2an^2 d + 8nh^2)l \tag{10}$$

$\square$

**Theorem I.3.** *(**FLOPS for adding** $k$ **layers vs.** $p$ **pause tokens**) Consider a $l$ layer decoder only model, with $h$ denoting the input embedding dimension and $d$ denoting the per attention head embedding dimension. Let $n$ be the length of initial prompt. Then, under the assumption that hidden embedding dimensions are much larger then the prompt sequence length and the appended pause tokens i.e. $d, h \gg n, p$; the additional FLOPS from $p$ pause tokens is less than that from $k$ additional layers of transformer if $n > pl/k$.*

*Proof.* From Lemma I.2 we have that increase in FLOPS due to $k$ additional transformer layer is give by:

$$F_{\Delta l = k} = F(n, h, a, l + k) - F(n, h, a, l)$$
$$F_{\Delta l = k} = (4nadh + 2an^2 d + 8nh^2)k \tag{11}$$

Similarly, increase in FLOPS due to $p$ pause tokens is given by:

$$F_{\Delta n = p} = F(n + p, h, a, l) - F(n, h, a, l)$$
$$F_{\Delta n = p} = (4padh + 2ad((n + p)^2 - n^2) + 8ph^2)l$$
$$F_{\Delta n = p} = (4adh + 2ad(2n + p) + 8h^2)pl \tag{12}$$

Now,

$$F_{\Delta n = p} < F_{\Delta l = k}$$
$$\implies (2adh + ad(2n + p) + 4h^2)pl < (2adh + and + 4h^2)nk$$
$$\implies n > \frac{(2adh + ad(2n + p) + 4h^2)pl}{(2adh + and + 4h^2)k}$$
$$\implies n > \frac{pl}{k} \qquad [\text{assuming } h, d \gg n, p] \tag{13}$$

$\square$

Next, we derive how many attention heads can be added, to be FLOP-equivalent to adding $p$ pause tokens. Note that in the standard transformer implementation, increasing the attention heads decreases the per attention head embedding dimension (i.e. $d = h/a$). In contrast, adding pause tokens, increases the number of attention computations while keeping the per-attention-head dimension fixed. Thus for a fair comparison, we consider a setting where we increase the number of attention heads, while keeping the per-attention-head dimension fixed. Specifically, we consider the case where per attention head embedding dimension is fixed to be the same of input embedding (i.e. $d = h$).

**Theorem I.4.** *(FLOPS for adding $k$ **attention heads** vs $p$ **pause tokens**) Consider a decoder only language model, with the per-attention-head embedding dimension $d$, fixed to be same as the input embedding dimension $h$. Let $n$ be the length of initial prompt. Then under the assumption that hidden embedding dimension is much larger then the prompt sequence length and the appended pause tokens i.e. $d, h \gg n, p$; the additional FLOPS from $p$ pause tokens is less than that from $k$ additional attention head, if $n > \frac{(a+2)p}{k}$.*

*Proof.* From Lemma I.2, we have:

$$F(n, h, d, a, l) = (4anh^2 + 2an^2h + 8nh^2)l, \text{ where } d = h. \tag{14}$$

Now, increase in FLOPS due to $k$ additional attention head is given by:

$$F_{\Delta a = k} = F(n, h, a + k, l) - F(n, h, a, l)$$
$$F_{\Delta a = k} = (4knh^2 + 2kn^2h)l \tag{15}$$

Similarly, increase in FLOPS due to $p$ pause tokens is given by:

$$F_{\Delta n = p} = F(n + P, h, a, l) - F(n, h, a, l)$$
$$F_{\Delta n = p} = (4aph^2 + 2ah((n + p)^2 - n^2) + 8ph^2)l$$
$$F_{\Delta n = p} = (4ah + 2a(2n + p) + 8h)phl \tag{16}$$

Therefore, we have, for:

$$F_{\Delta n = p} < F_{\Delta a = 1}$$
$$\implies (4ah + 4an + 2ap + 8h)hpl < (4nh^2 + 2n^2h)lk$$
$$\implies ((2a + 4)h + 2an + ap)p < (2h + n)kn$$
$$\implies n > \frac{(2a + 4)h + 2an + ap}{(2h + n)k} \cdot p$$
$$\implies n > \frac{(a + 2)p}{k}, \qquad [\text{assuming } h, d \gg n, p] \tag{17}$$

$\square$

## J  THEORETICAL INTUITION

This section formalizes a broad class of problems where appending pause tokens during inference can enhance expressivity and thus be helpful. Our formalization identifies two core insights:

1. *Pause tokens can be critical to solve tasks that require a large number of independent parallel computations that exceed the number of input tokens.* For example, consider a task where the input is a sequence of $L$ numbers $v_1, v_2, \ldots, v_L$, and the target is a polynomial of the form $(v_1 + v_2) \cdot (v_1 + v_3) \ldots (v_5 + v_2)$. If the number of addition operations required ($N$) scales much larger than the total number of input tokens ($L$), (and so $N = \omega(L)$) we argue that (under some natural capacity constraints), standard inference fails as it is bottlenecked in terms of its "implementational capacity": it can conduct only $O(L)$ operations in parallel. Pause-inference however is relieved of this bottleneck.

2. *The attention-feedforward block in any layer has "untapped" representational capacity — that is independent of the input length — which pause-inference taps into.* Specifically, note that the attention-feedforward block can implement many different operations, one for each intermediate vector it generates at each positional index. But crucially, the number of possible such operations (say, $K$) scales with the parameter count of the block. This quantity is independent of — and in practice, is much larger than — the input sequence length. Unfortunately, standard inference can only help realize at most $L$ such operations (where $L \ll K$), while pause-inference can tap into $K$ different such operations.

Combining the above two insights, our main result stated informally is that, given a fixed (2-layer) Transformer architecture, (a) if the underlying task requires $N$ parallel operations, where $N$ exceeds the number of input tokens $L$, and (b) as long as $N$ is not much larger than the parameter count $K$ of the attention-feedforward block, pause-inference can solve tasks that standard inference cannot.

We formalize the above insights in the form of assumptions stated in an abstract setting (in order to be as general as possible). We emphasize that the crux of our argument lies within these assumptions themselves, rather than the proof of our theorem. Thus, our main result here should be viewed as identifying precisely what assumptions are required for pause-inference to help.

## J.1 Underlying task

We consider an abstract set of tasks that require a first step that involve multiple parallel operations, following by a simple aggregation step to arrive at the solution:

**Assumption J.1.** *(structure of the underlying task) Given the vocabulary space $\mathcal{V}$, let $\circ$ be a generic 2-ary operator on $\mathcal{V}$. For an input sequence length $L$, consider a corresponding function class $\mathcal{F}_L$ that corresponds to all functions $f : \mathcal{V}^L \mapsto \mathcal{V}$ that require applying $N \circ$ operations independently following by a generic aggregation operation $g_{\mathrm{aggr}} : \mathcal{V}^N \to \mathcal{V}$:*

$$\mathcal{F}_L = \left\{ f : \mathcal{V}^L \mapsto \mathcal{V} \,\middle|\, \exists i_1, \ldots i_N, j_1, \ldots j_N \in [1, L], \right. \tag{18}$$

$$\left. f(v_1, v_2, \ldots, v_L) = g_{aggr}\big( \underbrace{(v_{i_1} \circ v_{j_1}), (v_{i_2} \circ v_{j_2}), \ldots, (v_{i_N} \circ v_{j_N})}_{N\,independent\,\circ\,operations.} \big) \right\}. \tag{19}$$

**Examples.** This structure covers a broad range of examples.

- As a simple mathematical example, this covers learning polynomials of the form $(x_1 + x_2) \cdot (x_3 + x_4) \cdot (x_1 + x_3)$.
- As a natural language example, consider a multi-choice question-answer task with $C$ choices given along with $E$ pieces of evidence in the context. One can then imagine that each $v_i$ corresponds to a piece of evidence, and each $v_j$ a choice. We then require $N = C \cdot E$ operations that compare each of the given choices against each of the given pieces of evidence. A final aggregation step would select the choice for which there exists a piece of evidence the most confidently corroborates the choice.

## J.2 (Tight) Upper bounds on the Transformer capacity

If the attention-feedforward module had, say, an infinite or exceedingly large capacity, the Transformer would be able to trivially express the solution to any task. It is only when these modules have finite capacity — as they do in practice — that we expect additional operations introduced by the pause tokens to be helpful. Correspondingly, we state this intuitive assumption as multiple "tight upper bound" assumptions. Each assumption states that the modules in a Transformer *can* represent objects of a certain complexity, but none any more complex than that.

Our first such assumption is in how much information can be represented by each intermediate vector. Specifically, we assume that each vector can precisely capture one token in $\mathcal{V}$ along with the positional index of the token (akin to positional embeddings injected into each token in practice). The precise form by which this information is represented as a vector is abstracted away for our discussion (e.g., it may be in one-hot form). Also note that our argument can be extended to settings where each intermediate vector could potentially represent more tokens, we discuss this at the end of the section.

**Assumption J.2.** *(information bandwidth limit of intermediate Transformer operations) We assume that the $i$'th intermediate vector in any given layer can be represented as $(u_i, i) \in \mathcal{V} \times \mathbb{N}$.*

Next, we assume a finite limit on the class of functions that each intermediate Transformer operation can represent. To state this, let $\mathbf{u} = ((u_1, 1), (u_2, 2), \ldots, (u_L, L))$ be the outputs of an intermediate

layer (corresponding to a $L$-length input sequence). For convenience, ignoring the residual and layer-norm blocks, let the $i$'th output of the next layer be expressed as:

$$\phi_{\text{FF}}(\phi_{\text{Attn}}(\mathbf{u}, \mathbf{u}, (u_i, i))) = (u'_i, i) \tag{20}$$

where the first two arguments represent the keys and values, and the third argument the query for the $i$'th intermediate operation in the considered layer. Note that $\phi_{\text{FF}}$ and $\phi_{\text{Attn}}$ are parameterized modules that can implement a finite set of functions. We assume what this set of functions consists of:

**Assumption J.3.** *(representational limits of intermediate Transformer operations) We assume that for each index $i \in \mathbb{N}$, $\phi_{\text{FF}}(\phi_{\text{Attn}}(\cdot, \cdot, (\cdot, i))$ can represent exactly one of two types of functions:*

- *A **single** $\circ$ operation. Specifically, we assume the self-attention operation can **select two indices** as $\phi_{\text{Attn}}(\mathbf{u}, \mathbf{u}, (u_i, i)) = (u_{\nu(i)}, u_{\nu'(i)})$, where $\nu, \nu' : \mathbb{N} \to \mathbb{N}$ come from some finite set of "index-selecting" functions $\mathcal{P}$. We then assume $\phi_{\text{FF}}$ can **implement** $\circ$ as $\phi_{\text{FF}}(u_{\nu(i)}, u_{\nu'(i)}) = u_{\nu(i)} \circ u_{\nu'(i)}$.*

- *The aggregating function $g_{\text{agg}}$ as $\phi_{\text{FF}}(\phi_{\text{Attn}}(\mathbf{u}, \mathbf{u}, (u_i, i))) = (g_{\text{aggr}}(u_1, \ldots, u_L), i)$.*

We explain why the above sub-assumptions are both reasonable and can hold simultaneously. First, we argue why it is reasonable that each intermediate Transformer operation can implement a limited number of $\circ$ operations, but not any more. Assume that the model needs to represent $u'_{35} = u_1 \circ u_3$. This requires the model to pay attention to the query's positional index 35, and then select the values at two different positional indices 1 and 3. Selecting these two values can be implemented by 2 self-attention heads operating independently. A subsequent feedforward network can then operate on a concatenated input $(u_1, u_3)$. Crucially, implementing any further $\circ$ operations, would require more attention heads. Thus, it is reasonable to assume a (tight) limit on the number of $\circ$ attention operations.

Note that the above assumption can simultaneously hold with the assumption that the Transformer operation can implement the aggregation function $g_{\text{aggr}}$. This is because $g_{\text{aggr}}$ does not require preferentially selecting any positional indices: all inputs are aggregated equally. Thus, we only need a single self-attention head that applies equal weight to all the values.

The last and arguably most insightful assumption stipulates a tight upper limit on what the self-attention module can represent. Specifically, observe that above we assume that the self-attention module can implement a finite set of functions $\mathcal{P}$, which help select "indices" $\nu(i)$ and $\nu'(i)$. We assume that there is a limit to this set of index-selecting functions $\mathcal{P}$. *Our key insight is that this limit is purely determined by the parameter count of the module, and is therefore independent of the input-sequence-length, $L$.* In practice, the size of the set $\mathcal{P}$ is much larger than the number of tokens $L$, and thus corresponds to the untapped capacity in the model. This capacity is however bottlenecked by the input length $L$, which determines how many operations in $\mathcal{P}$ are executed in standard inference.

**Assumption J.4.** *(the "untapped" capacity of self-attention operation is independent of input length) We assume that for some $K \gg L$, the self-attention module in each layer has at least $2K \log K$ parameters, and can hence implement the index-selecting functions $\nu, \nu' : [K] \to [K]$ to be any of the $K^K$ many mappings possible. In other words, $\mathcal{P} = \{1, 2, \ldots, K\}^{\{1,2,\ldots,K\}}$.*

From our assumptions, we derive the following result on what pause-inference can implement, which standard inference cannot, given a 2-layer Transformer (we discuss extensions to larger architectures in a subsequent remark):

**Theorem J.1.** *Under Assumptions J.1, J.2, J.3 and J.4, standard inference on a 2-layer Transformer with no pause tokens can only represent the function class $\mathcal{F}_N$ for $N \le L$, where $N$ denotes the number of parallel operations required by the function class, and $L$ denotes the length of the input sequence. In contrast, a 2-layer Transformer with $N - L$ appended dummy tokens can represent the function class $\mathcal{F}_N$ for any large $N \ge L$ as long as $N \le K$, where $K$ scales with the parameter count of the self-attention module as in Assumption J.4.*

The key insight is that the self-attention module has the representational capacity to implement $K$ different $\circ$ operations, a capacity that is independent of, and much larger than the number of

input tokens. However the standard Transformer that sees only $L$ tokens, only allows the model to realize $L$ of these. We hope this serves as a preliminary way of formalization the notion of the "implementation capacity" of a Transformer vs. the notion of raw parameter capacity.

*Proof.* Under Assumption J.2 that each intermediate vector can only represent one token from $\mathcal{V}$, the first layer in the Transformer will have to implement all the $N \circ$ operations to represent any given $f \in \mathcal{F}_L$. Therefore, at the $k$'th index, the model would have to instantiate $\nu(i) = i_k$ and $\nu'(i) = j_k$ (as defined in Assumption J.3), so that the corresponding Transformer operation can compute $v_{i_k} \circ v_{j_k}$. However, by Assumption J.3 (which states that each intermediate Transformer operation can implement only one $\circ$ operation) and by how standard inference in a Transformer is defined (which allows only as many operations as there are input tokens), the first layer can only compute at most $L$ many $\circ$ operations. Hence, a Transformer with standard inference can only represent $\mathcal{F}_N$ for $N \leq L$. On the other hand, from Assumption J.4, we know that the self-attention operator can implement as many as $K$ such operations, where $K \gg L$. Thus as long as $N \leq K$, with $N - L$ dummy tokens, the Transformer can implement $\mathcal{F}_N$ for $N > L$. □

**Remark J.2.** *(**breaking the information bandwidth assumption**) In Assumption J.2, recall that we assume that in each intermediate vector, we are able to communicate precisely one token from $\mathcal{V}$. If this assumption were to break, then standard inference would be able to implement a larger class of functions where $N > L$. However, it would still fall short of what pause inference can do. Specifically, imagine that each layer passes on its computed output, and also all computed outputs from the previous layer. Thus, if the intermediate vector can represent upto $N/L$ tokens, then the Transformer could divide the $N$ required $\circ$ operations over $N/L$ many layers, each layer performing $L$ operations in parallel. This would however be slower as it requires a series of $N/L$ computations. In contrast, pause-inference requires only $1$ set of parallel computations, and also a meagre information bandwidth of $1$ token per intermediate vector.*

**Remark J.3.** *(**recurrent depth**) An alternative way to exploit the untapped capacity of the attention-feedforward modules, is to repeat these operations sequentially along the depth of the model as done in Dehghani et al. (2019). This strategy would be helpful in tasks that require recursion. However, our class of problems do not involve recursion of any form. Thus, if these recurrent layers simply repeat the same $\circ$ operations over and over, we may not enjoy any advantages. However, one may argue that, perhaps repeating the same layers over and over somehow implements different operations in each repetition. In this case, we can make an argument similar to the previous remark. Specifically, to fit a task that requires $N$ parallel operations, we would need a model that has an information bandwidth of $N/L$, and applies a recurrence of $N/L$ layers corresponding to $N/L$ serial operations to compute the desired function. This should again be contrasted with pause-inference which requires only $1$ set of parallel computations, and also a meagre information bandwidth of $1$ token per intermediate vector.*

## K  ADDITIONAL RELATED WORK

**Input-only tokens.** The idea of using tokens that occur only as an input has found its use in various forms, most commonly as `<cls>` (Chang et al., 2023; Liu, 2019; Devlin et al., 2019), `<sep>` or `<mask>` in BERT (Devlin et al., 2019) and in a line of work on adding memory to transformers (Burtsev et al., 2020; Bulatov et al., 2022; Darcet et al., 2023).

**Chain-of-thought (CoT) prompting and role of intermediate tokens.** CoT prompting has been shown to significantly improve the reasoning capabilities of large language models (Wei et al., 2022; Nye et al., 2021; Lanchantin et al., 2023; Suzgun et al., 2023; Zelikman et al., 2022; Zhou et al., 2023; Wang et al., 2023b; Yao et al., 2023). Consequently, there has been a surge of interest in understanding the source of these CoT prompting gains. Feng et al. (2023); Merrill & Sabharwal (2023) theoretically argue that CoT aids by increasing the computational expressivity of the Transformer. Other empirical works (Turpin et al., 2023; Wang et al., 2023a; Madaan & Yazdanbakhsh, 2022) have shown that the generated intermediate reasoning steps can be unfaithful, not representing the model's true reasoning process. Wang et al. (2023a) empirically show that even incorrect reasoning steps can preserve $80\% - 90\%$ of the performance gains. However, Lanham et al. (2023) find that simply replacing the reasoning steps with filler tokens is unhelpful. As we argue, the model needs to be primed to process such tokens to help its computation.

**Lightweight finetuning techniques.** Pause-finetuning bears some resemblance to an orthogonal line of work on lightweight finetuning and ensembling techniques (Liu et al., 2022; Li & Liang, 2021; Lester et al., 2021; Hambardzumyan et al., 2021; Qin & Eisner, 2021; Logeswaran et al., 2020; Liu et al., 2021; Zhong et al., 2021; Schick & Schütze, 2021; Xue et al., 2022; Chang et al., 2023). Lightweight finetuning is concerned with parameter-efficient techniques that do not update the model's weights, and instead update a series of *multiple distinct* learnable tokens (prepended to the input). While pause-training uses a (single) learnable token too (appended to the input), the goal and effects are significantly different. First, pause-training is not intended for parameter-efficient finetuning. Infact, pause-training tunes slightly more parameters than standard finetuning. Next, in terms of the effect, while pause-training hopes to outperform standard finetuning as it is a less constrained technique, lightweight finetuning typically cannot, as it is a more constrained technique. Finally, note that pause-training crucially benefits from introducing the <pause> tokens during pretraining, while lightweight methods do not affect pretraining in any way.

**Adaptive compute.** In the literature, there are two major paradigms of adaptive compute. In the cascading paradigm, one selects between models of varying sizes (Jitkrittum et al., 2023; Kusupati et al., 2022; Devvrit et al., 2023). Another standard approach towards adaptive compute is layerwise adaptive compute within a single model, called early-exiting (Schuster et al., 2022; Schwartz et al., 2020; Eyzaguirre et al., 2021; Banino et al., 2021).

It is worth noting that while adaptiveness of the input token length (Xue et al., 2023) helps expand the (parallel) computational width, early-exit/layer-recurrence type of methods help expand the (serial) computational depth of the model. As formalized in Theorem J.1, the expanded computational width is critical in a range of problems.

