# OpenReview forum: "Think before you speak: Training Language Models With Pause Tokens"
_ICLR.cc/2024/Conference — ICLR 2024 poster_

### Official Review · Reviewer_cnas · 2023-10-30

**Soundness:** 3 good
**Presentation:** 3 good
**Contribution:** 3 good
**Rating:** 8
**Confidence:** 4

**Summary:**

This paper presents a method to inject pause tokens into language models during pretraining and fine-tuning, such that the language model can spend more tokens/compute before outputting the final answer. Specifically, pause tokens are randomly injected into sequence during pretraining and appended to questions during fine-tuning. Extensive experiments on several datasets demonstrate the robust gain induced by this method.

**Strengths:**

- The paper is well written and easy to follow.
- The method is novel and interesting.
- The experiments are extensive and demonstrate clear gains.

**Weaknesses:**

- It seems mysterious and problematic that sometimes more pause tokens can lead to worse performance.

**Questions:**

- How does one randomly inject pause token during pretraining? Could there be more than one pause token in a row during pretraining? Is there an upperbound on how many pause tokens can be in a row?
- Have you tried to prompt LLM with filler words that typically represent pause or thinking? Like "let me think a bit ...." etc?

---

> ### Author Response · Authors · 2023-11-22
> **Thank you! Our response:**
>
> We thank the reviewer for their time and positive feedback. We are happy to note that the reviewer found our work “novel and interesting” and appreciates the “extensive experiments and clear gains”. Below we address the concerns they raised.
>
> > It seems mysterious and problematic that sometimes more pause tokens can lead to worse performance.
>
> This is an interesting question, **but we believe that this is not mysterious if we view it from the fundamental lens of _overfitting_.** Recall that if a model is inundated with a lot of features, it can fail to generalize well as it is unable to pick up on the general patterns in those features. In our case, if we look at each intermediate vector input to some hidden layer as a feature, when we add $p$ pause tokens, we add $p$ new features. When we add an excessive number of pause tokens, the subsequent layer may suffer from overfitting. Of course, the value of $p$ for which overfitting starts to happen, must vary from task to task, depending on the task’s complexity.
>
> > How are pause tokens injected during pretraining
>
> We add pause tokens at uniformly randomly selected positions in input sequences. We then  trim the appended sequence back to the original length. We do this so as to have a fair comparison with standard pretraining without pause tokens: this way, we preserve the same computational cost during pretraining.
>
> > Could there be more than one pause token in a row during pretraining? Is there an upperbound on how many pause tokens can be in a row?
>
> Yes, there can be some cases  where many pause tokens come in a row. In fact, it is an interesting direction for future work to come up with better ways of adding pause tokens during pretraining, as the consecutive pause tokens can be more helpful. This is because at inference, we do add a consecutive chunk of pause tokens at the end of prompt.
>
> > Have you tried to prompt LLM with filler words that typically represent pause or thinking?
> What we tried was appending periods (“.”) as the pause tokens, as discussed in Section 4.3. We doubt that even prompting with words like “let me think a bit” will allow the model to leverage additional computational pathways within the transformer, as our work clearly shows that one needs to pretrain the model accordingly for the same.
>
> Thanks again for your time and for your feedback!

---

> > ### Comment · Reviewer_cnas · 2023-11-22
> >
> > Thank you for the response. I will maintain the current score.

---

> > > ### Author Response · Authors · 2023-11-22
> > > **Thank You**
> > >
> > > We thank the reviewer again for their time and valuable feedback, which helped us improve our work. Thank you for maintaining your score and assessing the work in a positive light.

---

### Official Review · Reviewer_zLe3 · 2023-11-01

**Soundness:** 3 good
**Presentation:** 4 excellent
**Contribution:** 1 poor
**Rating:** 3
**Confidence:** 4

**Summary:**

The paper presents and analyzes Pause Training - a technique for training transformer language models while inserting virtual "pause" tokens between real tokens. The model is then trained normally, except that the training loss does not apply to predictions at pause tokens, essentially letting the model do whatever it wants as long as it predicts the next true token correctly. The intent behind pause training is to let the model do additional computation ("thinking") before predicting the next real token. Authors consider several scenarios where pause tokens are introduced during pretraining or finetuning, for a range of 130M-1B model sizes, and evaluate several alternatives such as introducing extra real tokens (e.g. dots). Overall, authors observe the greatest gains from pause training when it is introduced at both pretraining  and finetuning stage, and find significantly smaller effect for finetuning only. The paper reports several additional experiments, including an ablation analysis of how many pause tokens are needed. Finally, authors draw several parallels to modern prompting and PEFT methods such as chain of thought and prompt tuning.

**Strengths:**

1. The paper presents a comprehensive experimental analysis of a (relatively simple) phenomena. Authors consider multiple downstream tasks, multiple model sizes, and tries several baselines. Overall, I believe that the quality of experiments is one of the stronger sides of this paper.

2. The paper focuses on a very simple algorithm, meaning that it can be easily implemented in most frameworks and libraries.

3. The paper is generally well written , both conceptual explanations and experiments were easy to follow. Discussions and parallels drawn were interesting to read.

**Weaknesses:**

My main problem with the paper is that it ignores several very similar works from the early years of Transformer.

Probably the closest idea to pause tokens is adaptive computation time (ACT) [1]. ACT was originally proposed for recurrent neural networks with the exact same motivation: to let the model "think" before generating a difficult token. The idea of ACT translates easily to Transformers, and many subsequent works in the last 7 years (e.g. [2,3]) use ACT for various transformer types, in both sequence and depth dimensions. If authors choose to also overview ACT in related work (and I encourage them to), please note that there are *many* more studies on adaptive computation time and I only referenced a few of them from the top of my head. Furthermore, unlike Pause Tokens, ACT offers a way of teaching model to automatically decide how many extra turns it needs to solve a problem.

With that in mind, I believe that a proper analysis of Pause Training should at least compare it to a temporal (regular) ACT for the same model with equivalent tuning budget.

* [1] https://arxiv.org/abs/1603.08983
* [2] https://arxiv.org/abs/1807.03819
* [3] https://arxiv.org/abs/2109.11745

Another related (but less similar) study is [4], that introduces additional learnable tokens for attention layers. The paper itself uses attention-only architecture, while subsequent works use the same idea for regular transformers, including a notable implementation[5]. To the best of my understanding, [4] proposes a mathematically similar idea that is likely to have similar effects to pause training, therefore there is no reason it shouldn't be a baseline for comparison. Additionally, [6-7] (and others) also propose to augment language models with attention to additional virtual tokens to improve language model quality. These other works pursue different goals from this paper, but, to the best of my knowledge, they are no less related than PEFT methods discussed on page 9.

* [4] https://arxiv.org/abs/1907.01470
* [5] https://github.com/lucidrains/x-transformers#augmenting-self-attention-with-persistent-memory
* [6] https://arxiv.org/abs/1911.00172
* [7] https://arxiv.org/abs/1612.04426


Overall, I believe that the paper presents an interesting idea which could offer a simpler and more efficient alternative to ACT-related methods, but it could also be inferior to them, and there is no way of knowing that from the experiments in the submitted draft. I would recommend authors to consider adressing the following questions for revised version of the paper:
- how does PT compare to ACT variants in terms of quality?
- how does PT compare to learned extra tokens (e.g. [4])?
- which approach is better at deciding where to put extra tokens?
- for modern language models, PT appears more efficient than a reasonably optimized ACT. How much faster is it in terms of training speed?
- which method is better for very few vs very many pause tokens?
(and similar)

Unfortunately, in the current form, I believe that answering those questions would require authors to reformulate too much of the paper content and message, and therefore neccesitate a full additional review. Therefore, I currently recommend rejection, but I encourage authors to resubmit a revised version once they can properly analyze how PT compares to earlier methods.
There are two circumstances in which I could raise the score within this review cycle:

1. authors prove that I misunderstood a significant chunk of the paper content and it is NOT in fact related to ACT family & others
2. authors prove that all of those questions can be answered trivially, without running many additional experiments or changing much of the paper

**Questions:**

On page 7, authors found that the optimal number of pause tokens depends on the specific task, e.g. GSM8k or SQuAD. Did you try to train the model to dynamically decide the number of pause tokens? Essentially, allowing model to predict  pause tokens at a certain penalty (like ACT) or spread a certain fixed budget of pause tokens between different positions (like CTC traning)?
The reason I'm asking is that, since the baselines are able to do the same thing, it would be curious to check which approach (PT or ACT or CTC) is better at deciding where to put the pause tokens.



Minor typo: authors use SquAD as a dataset name. To the best of my knowledge, the original paper[8] refers to this dataset as SQuAD, with uppercase Q.

[8] https://arxiv.org/abs/1606.05250

---

> ### Author Response · Authors · 2023-11-22
> **Thank you for missing references. We explain why these are completely orthogonal. (Part 1)**
>
> We thank the reviewer for their time and providing detailed feedback. The reviewer appreciated the “simplicity of our algorithm” and “comprehensive results” and raised some valid concerns about missing related work, which we address below.
>
> > My main problem with the paper is that it ignores several very similar works from the early years of Transformer.
>
> Thank you for pointing us to these lines of works, which we have now cited (Appendix K), along with a few other papers as well, which we had indeed missed in our original version. **However, we would like to strongly emphasize why our work is orthogonal to (and complements) adaptive compute approaches, and why it is unreasonable to compare against them as baselines:**
>
> ## Comparison with Adaptive Computation Time (ACT):
>
> 1.  **Ours is an extremely simple change that appends the same number of dummy tokens to all inputs, without adapting to the input or cleverly selecting any positions.** (during pre-training we randomly insert these tokens). In contrast, ACT [3][4][5] and related methods enjoy heavily-perfected adaptive machinery such as trained gating, halting, probing and routing mechanisms that allow them to adaptively allocate compute based on the input (and the model’s confidence etc.,). **Our current exploration is not concerned with such complex input-adaptiveness in any way.** The core question we ask is a much simpler one as you note — does appending dummy tokens even do anything? — for which we surprisingly find positive evidence. This, we believe is a significant novel finding in its own right. While we analyzed some basic varying of inference-time compute (Section 5.2) for the sake of ablations, we note that these again are done independent of the input i.e., we report accuracy, appending the same $p$ tokens to all inputs, and vary $p$ universally).
>
> **If** one compelled us to view this as comparable to adaptive compute techniques, we still argue that this is orthogonal and if at all, complementary:
>
> 2. A hypothetical pause-token-based adaptive compute technique would fall under a completely different paradigm of flexible compute (by varying input-length), rather than varying depth or model-size (which is used in cascades [6] or early-exit models, including DACT-BERT[4] and the Universal Transformer [3] you pointed us to). Importantly, increasing the input-length increases the computational width of the Transformer (i.e., the number of parallel operations) which is orthogonal to increasing the computational depth via parameter count or recurrent layers. **We formalize the benefits of expanding the computational width in the now-added Appendix J, where we show that there are tasks that benefit from this, but cannot benefit from depth-expansion in existing ACT.** Consider for example, computing $f(v_1, v_2, ... v_{50} ) = (v_1+v_{32}) \cdot (v_3 + v_{44}) ... (v_{27}+v_{2})$ where $f$ involves say  $1000$ many parallel addition operations — here, under some natural conditions, repeating sequential operation like in Universal Transformers would be of no use, while adding pause tokens would help implement many of these parallel operations!
>
> 3. On that note, adding recurrent layers like in Universal Transformers, would add much more to the wall-clock time during inference (owing to the extra sequential computations) compared to pause tokens (which only add pause-token-specific computations in parallel to existing computations).
>
> 4. To further appreciate why adaptive compute techniques are orthogonal to ours, observe that our technique can be directly combined with existing techniques like cascades, early-exits etc., i.e., one may increase/decrease the number of pause tokens, and decide to exit from a lower/higher layer.
>
> 5. Besides, it is worth noting that even within the ACT literature, cascade approaches and early-exit approaches treat each other as different paradigms, and do not typically compare against each other as baselines.
>
> To reiterate, regardless of our technical points 2, 3, 4 and 5 above, we believe that solely for point 1, our contributions should not be treated as comparable to adaptive compute time (ACT), nor do we claim to provide a competitive ACT-esque approach. **What our work explores is a fundamental question about a simple change to the Transformer.**  We agree that ACT is an important line of work that needs to be cited and discussed (Appendix K). **Unfortunately, we believe it is too reductive to reject our paper on the basis that we do not compare numbers with ACT algorithms that enjoy specialized machinery and are tangentially related.**

---

> ### Author Response · Authors · 2023-11-22
> **Thank you for missing references. We explain why these are completely orthogonal. (Part 2)**
>
> ## Comparison with Memory tokens
>
> Again, we are grateful for these excellent pointers. We had unfortunately missed this, but we have fixed this now (see Appendix K, where we have also added more works outside of your review).
>
> **We explain why the related Transformer works, which may appear superficially similar, are orthogonal to us.** Conceptually, ideas like Persistent Memory [7], RNN-cache [8] and kNN-LM [9] rely on introducing “global information” into the model. This:
>
>  - introduces significant number of new token parameters (e.g., the Persistent Memory tokens require 1024 global vectors in each layer, so roughly O(10M) new parameters).
>
>  - These vectors act only as keys and values, but not as queries.
>
>  - These vectors do not depend on the inputs.
>
>
> In contrast, pause tokens can be thought of as providing “local” information to the model by:
>
> - Introducing virtually no parameters besides a handful of parameters for the *single* pause token, (precisely 1 new vector, so O(1k) parameters)
>
> - Which subsequently generate intermediate vectors that act as keys, values **and queries**
>
> - And due to the above important difference, the new intermediate vectors depend on the input.
>
> ===
>
> ## Other questions:
>
> **Determining the number of pause tokens**: We determined the number of pause tokens by simply choosing between 10 or 50. We believe there is value in making the first step in the “pause tokens” direction simple so we can be confident that this core idea itself is promising.
> Complementing this with more sophisticated ACT-type techniques is an important direction for future work!
>
> ## Conclusion
>
> In summary, we completely agree with you that there are important related lines of work which we should have cited (and which we now have, please see Appendix K. We will reorganize the main paper to prominently include this discussion in future versions.).
>
> **We believe our work can complement and inspire new ACT approaches in the future. However, we hope you agree that reducing this paper to a new ACT-like approach [that should compete with other ACT approaches], and rejecting it on those grounds, is not well-warranted. We are afraid this characterization is unfortunately not accurate (as our approach does not claim to be, nor is as sophisticated as, an ACT approach) and unfortunately too reductive (as it ignores the value in our fundamental finding that adding a simple change can be helpful). We sincerely hope you will reconsider your score for our work in light of this discussion, and we once again thank you for pointing these important lines of work to us which we've now discussed in detail in the paper.**
>
>
> [1] Jason Wei, Xuezhi Wang, Dale Schuurmans, Maarten Bosma, Brian Ichter, Fei Xia, Ed H. Chi, Quoc V. Le, and Denny Zhou. Chain-of-thought prompting elicits reasoning in large language models. In NeurIPS, 2022.
> [2] Maxwell Nye, Anders Johan Andreassen, Guy Gur-Ari, Henryk Michalewski, Jacob Austin, David Bieber, David Dohan, Aitor Lewkowycz, Maarten Bosma, David Luan, Charles Sutton, and Augustus Odena. Show your work: Scratchpads for intermediate computation with language models, 2021.
> [3] Mostafa Dehghani, Stephan Gouws, Oriol Vinyals, Jakob Uszkoreit, and Łukasz Kaiser.  Universal transformers, 2019.
> [4] Cristobal Eyzaguirre, Felipe del Rio, Vladimir Araujo, and Alvaro Soto. Dact-bert: Differentiable adaptive computation time for an efficient bert inference, 2021.
> [5] Alex Graves. Adaptive computation time for recurrent neural networks, 2017.
> [6] Wittawat Jitkrittum, Neha Gupta, Aditya Krishna Menon, Harikrishna Narasimhan, Ankit Singh Rawat, and Sanjiv Kumar. When does confidence-based cascade deferral suffice?, 2023.
> [7] Sainbayar Sukhbaatar, Edouard Grave, Guillaume Lample, Herve Jegou, and Armand Joulin. Augmenting self-attention with persistent memory, 2019.
> [8] Edouard Grave, Armand Joulin, and Nicolas Usunier. Improving neural language models with a continuous cache, 2016.
> [9] Urvashi Khandelwal, Omer Levy, Dan Jurafsky, Luke Zettlemoyer, and Mike Lewis. Generalization through memorization: Nearest neighbor language models, 2020.

---

> > ### Author Response · Authors · 2023-11-22
> >
> > Dear reviewer,
> >
> > Since today is the last day of author-reviewer discussion period, if you have any quick follow-up questions we are happy to resolve them. We apologize for the delay in posting our responses since it took a while to  incorporate your feedback. Thanks once again for your time and valuable feedback.
> >
> > Authors

---

### Official Review · Reviewer_orpq · 2023-11-02

**Soundness:** 3 good
**Presentation:** 3 good
**Contribution:** 2 fair
**Rating:** 3
**Confidence:** 4

**Summary:**

This paper introduces an approach to the training and inference of language models by incorporating a concept of learnable "pause tokens." Language models generate responses token-by-token, where each subsequent token is influenced by the preceding ones. However, this research explores the impact of allowing the model to manipulate more hidden vectors by adding a sequence of learnable "pause" tokens to the input prefix before outputting the answer. The key idea is to allow the model extra computation time before committing to an answer. Experiments were conducted on models with different parameters and various tasks, which suggest that incorporating delays during inference time shows gains when the model is pre-trained and fine-tuned with these delays. Notably, significant gains were observed in tasks like question-answering on the SQuAD dataset, CommonSenseQA, and reasoning tasks.

**Strengths:**

1. This paper offers an intriguing exploration into the behavior of language models by increasing computation at the level of hidden states, rather than at the text level, like Chain-of-Thought. It presents the concept of the language model "thinking" through the generation of intermediate hidden states.
2. The introduction of a "pause token" is a novel approach that enables this deeper computational process to enhance the model’s performance on various tasks. There are a few useful observations. One is that pause tokens are necessary for both pre-training and fine-tuning stages.
3. The comprehensive experimentation involving various models and tasks contributes to the robustness of the paper. A strong aspect of the paper is its detailed ablation studies, which effectively dissect the influence of the "pause" token, thereby providing valuable insights into delayed next-token prediction.

**Weaknesses:**

1. Regardless of the empirical gains, we need more theoretical insights into why and how "pause tokens" work during pre-training and fine-tuning. There is not enough motivation behind this trick. We need to understand why we need to "delay a model's answer generation." There are a few intuitions, but are not well-articulated and convincing enough. The reason to answer this question is necessary because the community can benefit if the pause tokens are so important to replace normal autoregressive LLMs.
2. Adding the "pause token" brings new challenges and headaches. A limitation of the "pause token" approach lies in its necessity to be integrated into both the pre-training and fine-tuning phases, potentially hindering its broader applicability. The introduction of the "pause" token consumes a significant portion of the model’s input window space—approximately 10% of the sequence length—resulting in a loss of some training data information. Additionally, the number of "pause" tokens acts as a hyperparameter, exerting a considerable influence on task performance. This number is not consistent but varies across different tasks, adding a layer of complexity and variability to the model's application and performance.
3. The introduction of pause tokens undoubtedly adds a computational overhead to the training and inference processes. The paper could benefit from a more detailed analysis of this trade-off, especially in terms of computational resources and time. Assessing the scalability of this approach in larger models or in resource-constrained environments would be beneficial. Providing benchmarks on computational costs and suggesting ways to optimize this aspect could enhance the practicality of the approach.

**Questions:**

1. How do you determine the number of “pause” tokens?
2. How is the generalization ability of “pause” tokens, for example, pretraining with “pause” tokens and inference on tasks without fine-tuning? Are these tokens tasks specific?
3. The title doesn't convey any meaningful information, "Think before you speak" is too casual and just for eye-catching, and "Pause tokens" are not obviously justified in the title.

---

> ### Author Response · Authors · 2023-11-22
> **Added formal theoretical intuition and formal compute-efficiency analysis**
>
> Thank you for your well-articulated and precise criticisms of the paper. We have made significant additions to the paper based on your criticism as explained below.
>
> ## Theoretical intuition behind Pause tokens (added in Section J)
> >  There is not enough motivation behind this trick. We need to understand why we need to "delay a model's answer generation." There are a few intuitions but are not well-articulated and convincing enough.
>
> We completely agree with the concern that the intuition was originally fuzzy. To address this, **we have now added a rigorous and formal analysis of when and why pause tokens can help – please see Appendix J, leading to Theorem J.1.** To summarize for your convenience, we make two formal insights:
>
> 1. We formally show that pause tokens are helpful in problems that require a number of parallel operations that are much larger than the number of input tokens. For example, assume the model is given as input a sequence of $L$ numbers $v_1, v_2, ... v_L$, and it must compute a polynomial $(v_1 + v_2) \cdot (v_3 + v_1) ... (v_4 + v_8)$, that involves $N$ parallel additions, where $N \gg L$. A standard transformer can only implement $L$ operations in parallel, while appending pause tokens can allow implementing all $N$ operations.
> We argue that a similar case can arise in natural language settings e.g., consider an evidence-based multiple-choice question task with $C$ choices and $E$ pieces of evidence. Solving this would require $C \cdot E$ operations that individually corroborate each choice against each piece of evidence. But $C \cdot E$ scales much higher than the input sequence length which would scale as $O(C + E)$.
>
> 2. As another key insight, we argue that the self-attention/feedforward module has an “untapped” representational capacity that grows with its parameter count, say K. Importantly, this quantity is independent of the input length L. While standard inference only allows this module to realize L different operations, appending O(K) pause tokens can help realize as many K different operations.
>
> ## Challenges with Pause Tokens
>
> > Need to incorporate pause tokens during pretraining:
>
> While we agree that this is a hindrance, **this is in fact the key discovery of our exploratory paper which was hitherto unknown**. Further, some recent papers like [1] also argue that post hoc learning of chain-of-thought style reasoning may not work very well if only introduced in a post-hoc manner, as the default next token prediction pretraining objective (without learning to rely on the reasoning during training) does not model the same.
>
> > Pause tokens consume model’s input window space:
>
> There appears to be a misunderstanding here: we deliberately fixed the window length same as standard training for the sake of fairness of computational costs for both models. In fact, this indeed further highlights another interesting aspect about pause training – it is able to outperform standard training even while being at a disadvantage due to less number of “real” tokens seen!
>
>
> > Number of pause tokens as the hyper-parameter:
>
> We agree that this hyperparameter can be a nuisance.  However, we note that this was varied (and that too, over a grid of two values!) only in the downstream finetuning, and not during pre-training.
>
> [1] Merrill, W., & Sabharwal, A. (2023). The Expressive Power of Transformers with Chain of Thought. ArXiv, abs/2310.07923.

---

> ### Author Response · Authors · 2023-11-22
> **Added formal theoretical intuition and formal compute-efficiency analysis (part 2)**
>
> ## Assessing computational costs  (added in Section I)
> > adds a computational overhead to the training and inference processes… The paper could benefit from a more detailed analysis of this trade-off,
>
> You have raised some important practically-relevant questions regarding the compute-efficiency of pause-inference. **We have conducted an extensive analytical study of this in Appendix I (and Theorem I.3 and Theorem 1.4) in terms of FLOPS-efficiency and wall-clock-compute-efficiency.**  For your convenience, here are our main takeaways:
>
> - **Pause tokens are FLOPS-efficient compared to increasing parameter count:** If we were to add 10 pause tokens to our 1B model, how many extra FLOPS does it consume? Subsequently, if we allocate the same extra FLOPS budget to other ways of increasing the number of attention operations, how much accuracy gains would we see? Specifically, we consider (a) adding layers and (b) adding attention heads as ways to increase the number of attention operations. We find that for our architecture, adding 10 pause tokens at inference, over an initial prompt of length 100 tokens, becomes equivalent to (a) adding 2 layers (on top of 24 pre-existing layers) or (b) adding 3 attention heads per layer (in addition to 32 pre-existing heads). Both these approaches increase the parameter counts from 1B to (a) 1.1B and (b) 1.048B respectively, which is a modest increase in parameter count that cannot promise the gains we see in our 1B model with pause tokens.
>
> - **Pause tokens do not add extra serial compute:** Pause tokens are consumed in parallel with the prefix, with minimal wall-clock overhead. However, adding layers increases wall-clock time linearly with the number of layers. Similarly, other techniques like CoT introduce significant wall-clock overhead since they involve auto-regressively decoding each intermediate token (a wall-clock time of $pl$, where $p$ is the number of tokens and $l$ is the number of layers).
>
> ===
>
> ## Other questions
> > How do you determine the number of “pause” tokens?
>
> **This is a practically important question! In our current** exploratory work, we wanted to analyze the benefits of a simple no-frills approach of appending dummy tokens. Here, we simply choose between appending 10 or 50 tokens in the downstream task.   However, in the future, we could leverage the progress in the fields of adaptive computation and utilize some kind of confidence measure or gating units to introduce a variable number of pause tokens per input. Please note that we do not claim to propose any adaptive compute methods in the current submission.
>
> > How is the generalization ability of “pause” tokens, for example, pretraining with “pause” tokens and inference on tasks without fine-tuning? Are these tokens tasks specific?
>
> **To answer your question, we have added “zero shot evaluation” results in Appendix D**, where we do not finetune the model on the downstream task, and thus the pause tokens are untouched pretraining. Here, we compare the standard and the pause pretrained model (with 0, 10 and 50 pauses at inference). While on most of our tasks, our zero-shot values of both models are too low to be meaningful, we do observe that on datasets like GSM8k (mathematical reasoning) and PhysicalIQA (general understanding), pause tokens give some gains. We believe this provides preliminary evidence that these tokens are generalizable. Future work can help in improving this generalizability.
>
>
> > The title doesn't convey any meaningful information, "Think before you speak" is too casual and just for eye-catching, and "Pause tokens" are not obviously justified in the title.
>
> We will certainly rework the title in future versions.
>
> ## Conclusion
>
> We again thank the reviewer for their thoughtful and valid criticisms about the lack of (a) intuition and (b) compute efficiency analysis of pause tokens. **We believe that our updated paper and experiments fill these important gaps with key insights. We sincerely hope that you are able to re-assess this paper in light of these results in favor of acceptance.**

---

> > ### Author Response · Authors · 2023-11-22
> >
> > Dear reviewer,
> >
> > Since today is the last day of author-reviewer discussion period, if you have any quick follow-up questions we are happy to resolve them. We apologize for the delay in posting our responses since it took a while to incorporate your feedback. Thanks once again for your time and valuable feedback.
> >
> > Authors

---

### Official Review · Reviewer_SV1s · 2023-11-08

**Soundness:** 3 good
**Presentation:** 3 good
**Contribution:** 3 good
**Rating:** 8
**Confidence:** 3

**Summary:**

This paper shows that pretraining and tuning are necessary for consistent gains with pause tokens, which give transformers the ability to slightly decouple the amount of FLOPs from the number of tokens. The method inserts pause tokens at random positions during pretraining, and appends $M_{ft}$ and $M_{inf}$ pause tokens at fine-tuning and inference.

Experiments show that pause tokens improve performance for a variety of tasks.
Ablations show that, surprisingly, more pause tokens is not always better.

**Strengths:**

The paper successfully gets pause tokens working, which were previously thought not to work. The writing and presentation are clear, as are the experiments and results.

**Weaknesses:**

The paper is convincing and demonstrates some gains can be found via pause-token training. One weakness is that the method does not compare directly to Chain of Thought (CoT).

In contrast to pause tokens, which require both pretraining and fine-tuning, CoT is an inference-time only method that potentially requires extra human annotations. Another difference is that CoT has the ability to perform variable-length computation, as opposed to the fixed number of pause tokens added at inference time.

**Questions:**

Typos
1. Section 3.1: tokenresiding, the this
2. 4.3: standard-pretraing

Comments
1. A recent paper [1], that shows that extra scratch space improves the computational power of transformers, could help motivate this paper.

[1] Merrill, W., & Sabharwal, A. (2023). The Expressive Power of Transformers with Chain of Thought. ArXiv, abs/2310.07923.

---

> ### Author Response · Authors · 2023-11-22
> **Conceptual connections to CoT are interesting, but we explain why direct numerical comparisons are moot**
>
> We thank the reviewer for their time and feedback and for appreciating the experimental results. The reviewer raised a concern about comparison with CoT, which we address below.
>
> > Method does not compare directly with Chain of Thought
>
> The connections to CoT are indeed conceptually interesting, which we touched upon in the Related Work and Discussion sections. We completely agree that pause-training requires pre-training while CoT doesn’t — this is a great point of comparison. We have now added this to our discussion section. However, we wish to clarify that **a direct numerical comparison with CoT would be apples-to-oranges**, and the results are predictable a-priori. We would like to explain why below.
>
> In short, **CoT is a compute-hungry approach, while pause-training does not add (substantially) to wall-clock time:** each intermediate reasoning token in CoT is sequentially and auto-regressively decoded from the model. This naturally would give us high performance gains, but at a high compute cost. Formally, if there are $P$ intermediate CoT tokens and $L$ layers, CoT performs $P \cdot L$  **sequential** operations. Pause tokens on the other hand processes the P dummy identical tokens in **parallel**, with little excess in wall-clock time (depending on the number of parallel computational threads). Thus, by construction, pause-inference is computationally- and representationally-disadvantaged compared to CoT, rendering any numerical comparisons meaningless.
>
> Thus, as mentioned in the paper, we believe that in terms of accuracy, CoT is without question, an upper bound on pause-inference.  To address this discussion further, we have added relevant discussion to this in Appendix I.
>
> > CoT has the ability to perform variable-length computation
>
> This is an interesting aspect to consider! We agree that CoT can implicitly adapt its computation time to the input. But interestingly, on the flip side, **there is no explicit way to control the number of intermediate tokens in CoT, while this can easily be done for pause-inference.** In pause-inference, we can simply finetune different models with different numbers of pause-tokens and use whichever one we need depending on the “compute length” we require.  Alternatively, we also found a modest amount of robustness to varying the number of pause tokens at inference directly (while fine tuning with a fixed number of pause tokens) as shown in Figure 4c and 4d. (Note that this variation is not adaptive to the input though; it is done for all inputs.)
>
> Overall, we wish to emphasize that our work is aimed at highlighting how the simple approach of adding dummy tokens can surprisingly be helpful. While this can be thought of as a weaker form of CoT, we do not aim to propose an alternative to CoT and we believe the comparison to CoT is straightforward (in favor of CoT, if we care about accuracy).
>
>
> ===
> ### Other comments
> We thank the reviewer for pointing us to [1], which we have now cited in our work. The insights from this work are definitely relevant to us. It is worth noting that their analysis considers the CoT setting where the intermediate tokens are generated by the model itself, and fed back into the model. This enhances the “computational depth” of the model, which may be crucial to their theoretical claim. On the other hand, pause tokens enhance only the computational width of the model. In the now-added Appendix J, we formally argue why even this alone can be helpful in some tasks.
>
> We again thank the reviewer for their time and valuable feedback on the paper.
>
> [1] Merrill, W., & Sabharwal, A. (2023). The Expressive Power of Transformers with Chain of Thought. ArXiv, abs/2310.07923.

---

### Author Response · Authors · 2023-11-22
**Summary of major changes**

Thanks to all the reviewers for their constructive feedback. Taking their feedback into consideration, we have added the following important updates to our paper:


1. **An analytical study of inference-time compute-efficiency of pause tokens (Section I):** We analytically argue that pause tokens can be more FLOPS-efficient than other ways of adding attention operations, such as by adding layers and attention heads.
2. **A formal intuition for why pause token helps  (Section J):** We formally show that pause tokens can solve tasks that require a number of parallel operations that exceed the number of input tokens, while standard inference cannot.
3. **A more extensive comparison to related work (Section K):** We compare against adaptive-compute-time (ACT) line of work explaining why our work is orthogonal to (and is complementary to) ACT algorithms. We also argue how our work is orthogonal to other papers that introduce (a large number of) “memory tokens”  as keys or values as against (a single) pause token as key, value and query.

---

### Meta-Review · Area_Chair_1HnV · 2023-12-13

**Metareview:**

**Paper Summary:**

This paper proposes improving language models (LMs) by incorporating "pause tokens" during pretraining, finetuning, and inference. Pause tokens are inserted at random positions during training and appended to the input prefix during inference. The authors evaluated the approach on various tasks, including reasoning, question-answering, and general understanding, and found improved performance.

**Strengths:**

1. Strong Results: The authors conducted comprehensive experiments across multiple tasks and model sizes, demonstrating the effectiveness of pause tokens in various scenarios (SV1s, zLe3, cnas).

**Weaknesses:**

1. Scalability Concerns: While this paper has shown that the proposed technique works better for a 1B model compared to a 130M model, it is unclear whether such gains will still hold on larger models. A scaling law experiment where the gains are measured against varying parameter counts and training FLOPs would make this paper much stronger.
2. Lack of Theoretical Understanding: Despite empirical success, the paper lacks deeper theoretical insight into why and how pause tokens improve model performance. A more thorough motivation and explanation of the necessity and workings of pause tokens would strengthen the paper (orpq).
3. Complexity Concerns: The requirement to integrate pause tokens in both the pre-training and fine-tuning phases adds complexity (orpq).


**Decision:**

Based on the strong results and the potential to inspire future research in evaluating and understanding adaptive computation time techniques in LMs, I recommend accepting it as a poster.

**Justification For Why Not Higher Score:**

There is a lack of theoretical understanding of why adding pause tokens works. Besides, it is unclear whether this method works well at even larger scales, such as with tens of billions of parameters.

**Justification For Why Not Lower Score:**

Extensive experiments and strong empirical results.

---

### Decision · Program_Chairs · 2024-01-16

Accept (poster)